# Drug Combinations to Prevent Antimicrobial Resistance: Various Correlations and Laws, and Their Verifications, Thus Proposing Some Principles and a Preliminary Scheme

**DOI:** 10.3390/antibiotics11101279

**Published:** 2022-09-20

**Authors:** Houqin Yi, Ganjun Yuan, Shimin Li, Xuejie Xu, Yingying Guan, Li Zhang, Yu Yan

**Affiliations:** 1Biotechnological Engineering Center for Pharmaceutical Research and Development, Jiangxi Agricultural University, Nanchang 330045, China; 2Laboratory of Natural Medicine and Microbiological Drug, College of Bioscience and Bioengineering, Jiangxi Agricultural University, Nanchang 330045, China

**Keywords:** combination, antimicrobial resistance, selection index, collateral sensitivity, mutant prevention concentration, minimal inhibitory concentration, fractional inhibitory concentration index, stress factor

## Abstract

Antimicrobial resistance (AMR) has been a serious threat to human health, and combination therapy is proved to be an economic and effective strategy for fighting the resistance. However, the abuse of drug combinations conversely accelerates the spread of AMR. In our previous work, we concluded that the mutant selection indexes (SIs) of one agent against a specific bacterial strain are closely related to the proportions of two agents in a drug combination. To discover probable correlations, predictors and laws for further proposing feasible principles and schemes guiding the AMR-preventing practice, here, three aspects were further explored. First, the power function (*y* = a*x*^b^, a > 0) correlation between the SI (*y*) of one agent and the ratio (*x*) of two agents in a drug combination was further established based on the mathematical and statistical analyses for those experimental data, and two rules a_1_ × MIC_1_ = a_2_ × MIC_2_ and b_1_ + b_2_ = −1 were discovered from both equations of *y* = a_1_*x*^b1^ and *y* = a_2_*x*^b2^ respectively for two agents in drug combinations. Simultaneously, it was found that one agent with larger MPC alone for drug combinations showed greater potency for narrowing itself MSW and preventing the resistance. Second, a new concept, mutation-preventing selection index (MPSI) was proposed and used for evaluating the mutation-preventing potency difference of two agents in drug combination; a positive correlation between the MPSI and the mutant prevention concentration (MPC) or minimal inhibitory concentration (MIC) was subsequently established. Inspired by this, the significantly positive correlation, contrary to previous reports, between the MIC and the corresponding MPC of antimicrobial agents against pathogenic bacteria was established using 181 data pairs reported. These results together for the above three aspects indicate that the MPCs in alone and combination are very important indexes for drug combinations to predict the mutation-preventing effects and the trajectories of collateral sensitivity, and while the MPC of an agent can be roughly calculated from its corresponding MIC. Subsequently, the former conclusion was further verified and improved via antibiotic exposure to 43 groups designed as different drug concentrations and various proportions. The results further proposed that the C/MPC for the agent with larger proportion in drug combinations can be considered as a predictor and is the key to judge whether the resistance and the collateral sensitivity occur to two agents. Based on these above correlations, laws, and their verification experiments, some principles were proposed, and a diagram of the mutation-preventing effects and the resistant trajectories for drug combinations with different concentrations and ratios of two agents was presented. Simultaneously, the reciprocal of MPC alone (1/MPC), proposed as the stress factors of two agents in drug combinations, together with their SI in combination, is the key to predict the mutation-preventing potency and control the trajectories of collateral sensitivity. Finally, a preliminary scheme for antimicrobial combinations preventing AMR was further proposed for subsequent improvement research and clinic popularization, based on the above analyses and discussion. Moreover, some similar conclusions were speculated for triple or multiple drug combinations.

## 1. Introduction

Antimicrobial resistance (AMR) has been a serious threat to human health and economic development [1,2], and the COVID-19 pandemic accelerated this global problem [3]. Many strategies, such as the development of new antimicrobial agents [4], combination therapy [5], the optimal use of clinic antimicrobial agents [6], and non-antibiotic therapy [7], have been put forward for fighting or delaying resistance. Among them, combination therapy has been proved to be an economic and effective strategy for fighting the resistance, and many combinations have been explored for preventing AMR [5,8,9,10]. Many antimicrobial combinations have been explored for preventing AMR, and some of their effects on preventing the resistance are conflicting [5,10,11,12,13,14,15]. Even more, some combinations may result in high mutational frequencies [12,13], such as levofloxacin in combination with lower dose of colistin [12]. Recently, Liu et al. also revealed that bacterial tolerance promoted the evolution of resistance under combination treatments [16]. Therefore, it is urgent that scientists discover some regularity conclusions and put forward some proposals and applicable schemes for effectively guiding the practices of drug combinations preventing AMR, simultaneously avoiding the accelerated spread of AMR due to the abuse of drug combinations. 

To prevent AMR, hypotheses of mutant selection window (MSW) and mutant prevention concentration (MPC) were put forward by Zhao and Drlica [17]. According to these hypotheses, maintaining drug concentrations above MPC throughout therapy can severely restrict the acquisition of drug resistance. Based on these hypotheses, many related experiments have been performed for discovering probable parameters for predicting drug combination effects on the prevention of AMR or the selection of resistant mutant, and several MPC- or MSW-related parameters were proposed [18,19,20,21,22], such as MPC level (AAMPC), AUC_24_/MPC (area under the concentration-time curve over 24 h divided by the MPC) and *C*_max_/MPC (highest concentration divided by the MPC), and AUC_24_/MIC (AUC_24_ divided by the minimal inhibitory concentration) [23]. These results indicated that none of those predictors could be widely used to guide the practices of drug combinations preventing AMR.

To explore widely applicable predictors, some regularity conclusions on drug combinations preventing AMR were drawn in our previous works [24,25], and the important ones were listed as follows: (1) The MSWs and MPCs of one agent in combination are closely related to the proportions of two agents, and the lower the proportion of one agent in a drug combination is, the more likely the MSWs will be narrowed. Namely, the effect of a drug combination preventing AMR is closely related to the proportions of two agents, and different proportions will present different effects related to preventing AMR. (2) The MSWs of one antimicrobial agent can be narrowed or even closed by another in a drug combination whatever it is synergistic or not, although synergistic is better. Namely, many combinations have enough potential to prevent resistance, and the susceptibility of one agent can be enhanced by another even in an antagonistic combination [26]. Conversely, some improper combinations may result in high mutational frequencies [12], and which mainly depends on the proportions of two agents. These regularity conclusions can give reasonable explanations for the various results from drug combinations preventing AMR, including for contrary ones [11,12,13]. Simultaneously, they can also give reasonable explanations for the occurrence of collateral sensitivity [27] and for the fact that drug combinations possibly promote the transmission of resistance to a partner drug if the tolerance has already emerged to one drug [1,16]. These regularity conclusions also indicate that the abuse of drug combinations accelerates the spread of antimicrobial resistance.

As we concluded [24], the more remarkable the synergistic effect of two antimicrobial agents in a combination was, the more likely their MSWs were to be close each other, and the more difficultly the resistance emerged according to the hypotheses of MPC and MSW. Otherwise, the bacterial resistance to one agent would probably emerge when it narrows the MSW of another or prevents the resistance to another, especially inappropriate concentrations and proportions were administrated. However, it is difficult to acquire synergistic combinations [10], let alone that most combinations present different combinational effects on different pathogens. Furthermore, two antimicrobial agents in a drug combination usually present different pharmacokinetics parameters in vivo, and their proportions in blood and infectious tissue accordingly change. These must fluctuate or even invert the practical effects and increase the complexity and uncertainty of drug combinations preventing AMR. Therefore, these regularity conclusions are still unable to guide the practice of drug combinations preventing AMR, and only some regularity conclusions and proposals were provided in our previous works [24,28]. 

Here, the deeper analyses for our previous data [24] were performed for discovering probable correlations between the various indexes (such as MIC, MPC, fractional inhibitory concentration index (FICI) and mutant selection index (SI)), predictors, and laws. Then, the analysis conclusions were verified and improved through the communications with the practical effects preventing AMR of different proportional combinations after antibiotic exposure experiments. Based on these correlations and laws, and their verification experiments, some principles and a preliminary scheme that can guide the practice of drug combinations preventing AMR, together with some speculations for triple or multiple drug combinations and some proposals, were put forward for further experimental improvements and clinical trials.

## 2. Results

### 2.1. Correlation between the SI of One Agent and the Ratio of Two Agents, in a Drug Combination

The correlation between the SI (y) of one agent in a drug combination and the ratio (x) of two agents was further analyzed for the experimental data from three combinations roxithromycin/doxycycline (RM/DC), vancomycin/ofloxacin (VM/OX) and vancomycin/fosfomycin (VM/FF) in Table 1 and Table 2 reported by us [24], and 36 probable regression equations were established and presented in Table 1, together with their correlation coefficients (r). Simultaneously, their coefficients of determination (R^2^) were also calculated for comparing the goodness of fit of two equations established from the same data pairs. 

Using r-test in statistics, the results (Table 1) indicated that there are significant correlations between the SI of one agent and the ratio of two agents in most drug combinations, presenting the characteristics of power functions and/or logarithmic functions. In detail, only 5 of 36 data pairs showed no significant correlation, and simultaneously the r values of 3 of them were closed to the critical value r_0.975_ (5) of 0.754. Considering the possibility that the data deviations were caused by experimental errors, these above results concluded that two functions (1) and (2) can be established for the correlation between the SI (y) of one agent in a drug combination and the ratio (x) of two agents:*y* = a*x*^b^ (a > 0, *x* > 0)(1)
*y* = aln(*x*) + b (b > 0, *x* > 0)(2)
where *y* is the SI of one antimicrobial agent in a drug combination, and *x* is the ratio of another to this agent.

The R^2^ indicated as a whole that it was difficult to intuitively find an obvious difference between both two functions, and between the curves drawn by them. Therefore, communications between the mathematical characteristics and the related indexes of bacterial resistance, of the functions for two agents in a drug combination were further analyzed and shown in Table 2, for comparing which is the better one for predicting the mutant-preventing effect of two antimicrobial agents, and for predicting and controlling the trajectory of collateral sensitivity during the prevention from the AMR.

From Table 2, more information, laws, rules, and better fitting for some correlations can be obtained from function (1) than function (2). Therefore, it was concluded that the correlation between the SI of one agent in a drug combination and the ratio of two agents, presents the characteristics of power function *y* = a*x*^b^ (a > 0, *x* > 0). As the power function *y* = a*x*^b^ contains two unknowns a and b, a specific equation for the SIs of two drugs (A and B) changing with their ratios can be established by two data pairs consisting of the SI of one agent and the ratio of two agents in a drug combination. Namely, the a_1_ (or a_2_) and b_1_ (b_2_) of the specific equation *y* = a_1_*x*^b1^ (*y* = a_2_*x*^b2^) can be calculated from the experimental data of two SIs at two ratios of two agents, and it is better for two ratios of two agents to include 1 (1:1) and another (such as 8 or 1/8, or 4 or 1/4) since the curve of these power function must pass through the dot (1, a). After two equations *y* = a_1_*x*^b1^ and *y* = a_2_*x*^b2^ were respectively established for agents A and B, two rules (a_1_ × MIC_1_ =a_2_ × MIC_2_) and (b_1_ + b_2_ = −1) can be used for further verifying the two equations, and related applications. Therefrom, the SIs of one agent at any ratios of two agents in a drug combination can be calculated from *y* = a_1_*x*^b1^ and *y* = a_2_*x*^b2^, and the curves corresponding to both equations can be also obtained when need, without calculating SI for all combinational proportions.

Another, based on these correlations, laws and rules of drug combinations concluded from function (1) *y* = a*x*^b^ in Table 2, some typical curve outlines showing the representative correlations between the SI of one agent in drug combinations and the ratio of two agents can be drawn as Figure 1.

The other way round, the curves for agents A and B can also be roughly drawn according to the MICs, MPCs, SIs of two agents and the FICI of the combination, referring to Figure 1 and these correlations, laws and rules of drug combinations. The detail procedure was presented as follows: 

(1) The MICs and MPCs alone of two antimicrobial agents (A and B) are determined respectively using broth microdilution method and plate method with linear concentration decrease, and the FICI of combination A/B is tested using checkerboard method. 

(2) The MPCs in combination of two agents at the ratio of 1 are respectively determined for calculating their SIs (MPC in combination/MIC in alone, abbreviated as MPC_combination_/MIC_alone_). As the curves shown by function (1) *y* = a*x*^b^ must pass through the dot (1, a), at this moment, a_1_ and a_2_ are respectively equal to SI_1_ and SI_2_ and should meet the rule of a_1_ × MIC_1_=a_2_ × MIC_2_.

(3) The horizontal ordinate (x) representing the ratio of B/A (from left to right), and the longitudinal coordinates (y) representing the SI value (both sides) are drawn like Figure 1. Based on the results of above procedure (2), the dots (1, a_1_) and (1, a_2_) can be drawn for agent A and B, respectively. 

(4) According to the rule b_1_ + b_2_ = −1 and conclusions (11) and (12), the b_1_ (for A) and b_2_ (for B) are roughly estimated from the MPCs of both two agents and the regression equation (y = 28.831x − 27.831) between the ratio of b_larger_/b_smaller_ (b_1_/b_2_ or b_2_/b_1_) (x) and that of MPC_larger_ to MPC_smaller_ (MPC_1_/MPC_2_ or MPC_2_/MPC_1_) (y). According to both values, the curve outlines for agents A and B can be respectively drawn by simulating the curve characteristics of power function, referring to Figure 1.

(5) After above two curve outlines are drawn, a horizontal line y = 1 (SI = 1) (shown as blue dotted line in Figure 1) can be drawn for intuitively judging whether the MSWs are closed. Finally, the diagram of the SIs of two drugs changing with the ratios of two drugs in a drug combination is obtained. 

In addition to the above two methods, the function curves for the SIs of two drugs changing with the ratios of two agents can be also obtained by the fitting for power function from the experimental SI value at seven to nine ratios of two agents in drug combinations, according to similar method in Table 1. Therefrom, the SI of one agent at any ratios of two agents in a drug combination can be calculated or roughly estimated using above three methods, for predicting the mutant-preventing effect of two agents in drug combinations, and for predicting and controlling the trajectories of collateral sensitivity when need. 

### 2.2. Correlation between the MPSI and the MIC, MPC or SI Ratio of Two Agents in an Antimicrobial Combination

As the above concluded, the SI of one agent in a drug combination decreases along with the proportional increase of another agent. Simultaneously, the SI (MPC/MIC) reflects the closed degree of the MSW and is related to the mutation-preventing potency. Therefore, the difference of the maximum potency narrowing the MSW (namely, the potency difference in decreasing the SI), defined as mutation-preventing selection index (abbreviated as MPSI), of two agents in a ratio range (such as 1:64 to 64:1, or 1:64 to 8:1) of a drug combination, was first put forward for evaluating the difference of the potency preventing AMR and predicting or controlling the trajectory of collateral sensitivity.

According to the method described in Section 4.1.2, the MPSI determination is relatively complicated for two agents in a specific ratio range. However, it is easy for the MPSI to be calculated from both equations *y* = a_1_*x*^b1^ and *y* = a_2_*x*^b2^ after a_1_ (a_2_) and b_1_ (b_2_) were calculated, according to the method in Section 2.1, by the calculation from the experimental data pair (1, a) and another. Moreover, it is necessary to further explore whether there are more convenient and/or simpler predictors to roughly evaluate the MPSI of a drug combination in a specific ratio range of two agents, for predicting the difference in the mutation-preventing potency of two agents in probable experiments, trials and clinical practices.

To achieve this, the ratios for the MIC, MPC or SI (the ratio of MPC to MIC) alone of two antimicrobial agents in three combinations (roxithromycin/doxycycline, RM/DC; vancomycin/fosfomycin, VM/FF; vancomycin/ofloxacin, VM/OX) against three methicillin-resistant *Staphylococcus aureus* (MRSA) isolates were respectively calculated and are shown in Table 3. Correspondingly, the MPSI in the investigated ratio range from 8:1 to 1:8 of two agents, of three drug combinations against three MRSA isolates were also calculated according to the Formula (2) in Section 4.1.2, and shown in Table 3. 

Based on the data in Table 3, the correlation between the MIC, MPC, or SI ratio (*x*) and the MPSI (*y*), of two antimicrobial agents in three drug combinations against three MRSA isolates was analyzed. As the MPSI shows the potency difference narrowing the MSW of two agents A and B in drug combinations, the more the MPSI deviates from 1, the larger the potency difference narrowing the MSW of two agents. Namely, the larger the MPSI value when it is more than 1, the larger the potency for narrowing the MSW_A_ (the MSW of agent A); the smaller the MPSI value when which is less than 1, the larger the potency for narrowing the MSW_B_ (the MSW of agent B). Therefore, the reciprocals of the MPSIs were taken when the MPSI was less than 1, and correspondingly the reciprocals of the MIC, MPC, or SI ratios were also taken for the analysis of the correlation between the MIC, MPC, or SI ratio and the MPSI value in Table 3. The results indicated that there is a significantly (*P* < 0.01) positive correlation between the MPSI (*y*) and the MIC (μg/mL) or MPC (μg/mL) ratio (*x*), and six regression equations together with their correlation coefficients (*r*) were shown in Table 4. However, there is no significant (*P* > 0.01) correlation between the MPSI and the MIC (μM/L), the MPC (μM/L), or the SI ratio. 

The positive correlation between the MPSI and the MPC (μg/mL) or MIC (μg/mL) ratio indicated that the larger the MIC or MPC ratio (the larger divided by the smaller), the larger the MPSI value (the larger divided by the smaller), of two agents in drug combinations. Combined with the conclusion in Table 2, it indicated that the larger the difference between the MIC or MPC of two agents in a combination is, the more preferential the MSW of the agent with larger MIC or MPC is to be narrowed and even to be closed. That is to say, the agent with larger MIC or MPC has greater potency to keep its susceptibility to a certain pathogenic strain unchanged or even enhanced, and namely it has greater potency to fight the resistance from a certain pathogenic strain. Another, the *r* values in Table 4 indicated the MPC ratio is the best one correlated with the MPSI. This was further confirmed that the larger the difference between the MPC of two agents in a combination is, the more preferential the MSW of the agent with larger MPC is to be narrowed and even to be closed. It was also in accordance with the conclusion (11) shown in Table 2. Considering that the MICs of an antimicrobial agent are easy to calculate, the MIC ratio of two agents in drug combinations is probably an economic candidate in the practice of predicting the mutant-preventing potency. 

Moreover, as mentioned above, the MPSIs were set more than 1 when the correlations were established. This requires that the *y* values of Equations (1) to (6) be more than 1, so their *x* values should be more than 1.5 and 4.2. Namely, only when the MIC and MPC ratios of two agents in a drug combination are respectively more than 1.5 and 4.2, or less than 0.66 and 0.24, both two agents in drug combinations can present significant difference in mutant-preventing potency since the MPSIs are small and present a little fluctuation (Table 3) when the MIC or MPC difference of two agents is small. This was confirmed by the data pairs (4, 1.2848) and (0.533, 1.2848) respectively for the establishment of Equations (1) and (5). Therefore, it is better for two agents used as drug combinations to have enough difference in the MPCs, such as the MPC ratio larger than or equal to 4.2 (or less than or equal to 0.24), for accurately predicting and controlling the trajectories of collateral sensitivity.

As the correlation between the SI (*y*) of one agent in a drug combination and the ratio (*x*) of two agents has the characteristics of power function *y* = a*x*^b^, and the curve must pass through the dot (1, a). Simultaneously, the proportion 1:1, which *x* = 1, is the most common and representative proportion in drug combinations. Therefore, the MPSI of a drug combination at the ratio (1:1) of two agents, marked as MPSI (1:1) shown in Appendix A, is also calculated for reflecting the difference of the mutation-preventing potency of two agents with the combinational proportion 1:1. To explore whether the MPSI (namely, the tested MPSI in Appendix A) can be replaced by the tested or calculated MPSI (1:1) for evaluating the potency difference preventing AMR of two agents in a drug combination, the calculated MPSIs and calculated MPSIs (1:1) were respectively calculated from power function *y*= a*x*^b^ or logarithmic function *y* = aln(*x*) + b, and shown in Appendix A.

After the correlation analyses, it was surprisingly found that the MPSIs (1:1) (*y*) calculated from power function *y*= a*x*^b^ in Table 1 when *x* = 1 are approximately equal to the tested ones (*x*), with a linear equation *y* = 1.0398*x* and an *r* of 0.9972 (*P* < 0.01), and that there is significant (*P* < 0.01) correlation between the tested MPSIs (1:1) (*y*) and the tested MPSIs (*x*), with a linear equation *y* = 1.9392*x* and an *r* of 0.9672. These indicated that the tested or calculated MPSI (1:1) can replace the MPSI for effectively evaluating the potency difference preventing AMR of two agents in a drug combination, while the calculated MPSI (1:1) can be easily calculated after two equations *y* = a_1_*x*^b1^ and *y* = a_2_*x*^b2^ for two agents in a drug combination have been established using the data pair (1, a) and another, obtained from the experiment. Another, the significant correlation between the tested MPSI (*x*) and the calculated one (*y*), with a linear equation *y* = 1.5884*x* and an *r* of 0.9972 (*P* < 0.01), also indicated that the tested MPSI (namely, the MPSI in Table 3) can be roughly calculated from the established two equations *y* = a_1_*x*^b1^ and *y* = a_2_*x*^b2^ in a drug combination. However, five negative values were presented for nine MPSIs calculated from logarithmic function *y* = aln(*x*) + b in Table 1 when *x* = 1, and simultaneously it was found that there is no correlation between the tested MPSI (1:1) (*x*) and the calculated one (*y*) from logarithmic function, with a linear equation *y* = 0.4523*x* and an *r* of 0.4562 (*P* > 0.01). These further indicated power function *y*= a*x*^b^ is better than logarithmic one *y* = aln(*x*) + b to reflect the correlation between the SI value and the ratio of two agents in a drug combination.

Another, according to the experiment data obtained from the reported methods [24], all the MPC in combination of agent A is equal to that of agent B when the ratio of two agents is 1:1. Simultaneously, this MPC is the minimum MPC of agents A and B in a ratio range (alone to 1:1) according to the monotonic decreasing property of power function *y* = a*x*^b^ (a > 0, *x* > 0, mostly b < 0). Therefore, the calculation formula of the MPSI (1:1) can be simplified as the ratio of the MPC alone of agent A (${\mathrm{MPC}}_{A}^{\mathrm{Alone}}$) to that of agent B (${\mathrm{MPC}}_{B}^{\mathrm{Alone}}$) as following calculation Formula (1), according to Formula (2) in Section 4.1.2.
(1)$$\mathrm{MPSI}\;(1:1)={\mathrm{MPC}}_{A}^{\mathrm{Alone}}/{\mathrm{MPC}}_{B}^{\mathrm{Alone}}$$
where, the ${\mathrm{MPC}}_{A}^{\mathrm{Alone}}$ is the MPC (μg/mL) alone of agent A, and the ${\mathrm{MPC}}_{B}^{\mathrm{Alone}}$ (μg/mL) is that of agent B, in a drug combination A/B. 

As mentioned in Section 2.1, here Formula (1) further proved that the larger the difference between the MPC of two agents in a combination is, the more preferential the MSW of the agent with larger MPC is to be narrowed and even to be closed.

### 2.3. Correlation between the MPC and the MIC of an Antimicrobial Agent

Many papers [17,29,30,31] concluded that no obvious correlation between the MIC and the MPC of an antimicrobial agent was observed, and that the MPC couldn’t be predicted from the MIC. Here, the correlation between the MIC and the MPC was further analyzed, based on one hundred and eighty-one of data pairs (Appendix A) reported in fourteen papers [11,12,23,24,32,33,34,35,36,37,38,39,40,41]. The results were shown in Table 5, and six regression Equations (7) to (12) were established based on these data pairs using the mass concentration (μg/mL) and the molar concentration (μM/L), respectively. 

From Equations (7) and (10) in Table 5, the *r* values indicated that there is very significant (*P* < 0.01) positive correlation between the MICs and the corresponding MPCs of antimicrobial agents against pathogenic bacteria, especially when the mass concentration (μg/mL) was used for the correlation establishment. This indicated that the larger the MIC of an antimicrobial agent, the larger its MPC as a whole. To obtain more intuitive visual effects, the data pairs (MIC, MPC) were respectively transformed into the natural logarithm (log_10_) of the MIC and the MPC before their correlation analyses, and the results are shown in Figure 2; their regression Equations (9) and (12) are also shown in Table 5. This conclusion is contrary to the low correlation between them reported in previous publications [17,29,30,31], and the smaller number of samples is likely responsible for the low correlation concluded in previous reports, since the larger the number of samples, the closer the statistical result is to the essence of things. 

According to mathematical statistics, the larger the coefficient of determination (*R*^2^) is, the better the fit is. Comparing the *R*^2^ of Equations (7) and (8) indicated that Equation (8) *y =* 0.00006*x*^3^ − 0.07104*x*^2^ + 25.5154*x* was the better one for fitting the correlation between the MIC and corresponding MPC. However, there is no obvious difference between Equations (7) and (8) as a whole for the prediction of the MPC from the MIC of an antimicrobial agent, since both *R*^2^ values of Equations (7) and (8) are very close. Therefore, we can simultaneously use Equations (7) and (8), or other equations established like similar method, to quickly and complementarily predict the rough MPCs from the MIC as references when need. This was further verified by another forty-six data pairs (MIC, MPC) (Appendix A) from another five papers [42,43,44,45,46], with an acceptable probability of 91.3%. Conversely, the larger prediction accuracy further confirmed that the MPC of an antimicrobial agent can be roughly calculated from its corresponding MIC. Similarly, the rough MPC (μM/L) of an antimicrobial agent can be also calculated from its corresponding MIC (μM/L), using Equation (11) which is better than Equation (10) since there is obvious difference between Equations (10) and (11). However, the prediction reliability would be likely lower than Equations (7) and (8).

As the calculated MIC fluctuates within a reasonable range of the actual values, especially from 1/2 × to 2 × the actual one [47], the predicted MPC from the MIC would likely fluctuate in a certain range of the actual one, and sometimes only rough MPC can be calculated. However, it is very important for quickly selecting and screening the drugs of a drug combination depending on their MPCs calculated from the MICs. More directly, the determination process of MPC initial value can be omitted when the plate method with linear concentration decrease is used for the MPC determination. 

### 2.4. Verification and Improvement for Regularity Conclusion by Antibiotic Exposure Experiments

#### 2.4.1. Susceptibility Changes of MRSA to Antibiotics after Exposed to Reported Combinations

Combination RM/DC presents synergistic inhibitory effect to MRSA 01, but indifferent antimicrobial effect to MRSA 03. After respectively exposed to seven proportions (8:1, 4:1, 2:1, 1:1, 1:2, 1:4 and 1:8) of combination RM/DC, the susceptibilities of MRSA 01 and 03 respectively to RM and DC were determined, and the results were shown in Table 6.

Comparing the susceptibility changes of MRSA isolates to RM and DC after exposure with the SI in combination of RM and DC (Table 6), the susceptibility changes of MRSA 01 to RM or DC coincided with whether their corresponding MSW closed (SI ≤ 1) as a whole, only except that MRSA 01 was resistant to RM after exposed to the proportion (1:4) of combination RM/DC which the MSW (SI = 1.0) of RM against MRSA 01 was just closed. A similar situation also occurred in MRSA 03 to combination RM/DC, only that the susceptibility of MRSA 03, after exposed to the proportion (1:2) which the MSW (SI = 1.05) of RM against MRSA03 was also just closed, remained unchanged among three proportions (1:2), (1:4) and (1:8) of combination RM/DC. However, these two exceptions had no influence on the overall relationship between the susceptibility changes of pathogenic bacteria and to which the corresponding MSW of two agents closed or not, and this can be also confirmed by the corresponding change trend, shown in Figure 1 in our previous work [24], of MSWs closed along with the change in combinational proportion of two agents. Therefore, this indicated that the resistance would not happen if the MSWs of the agents in drug combinations to pathogenic bacteria were closed, which further confirmed the rationality for the hypotheses of MSW and MPC.

Another, observed from Table 6, the susceptibility of MRSA 01 to DC with a larger MPC remained unchanged for all proportions, while MRSA 01 is resistant to RM with a smaller MPC in five proportions after MRSA 01 exposed to seven proportions of combination RM/DC. Simultaneously, the susceptibility of MRSA 03 to RM with a larger MPC in the combination RM/DC remained unchanged or enhanced for all proportions, while that to DC with a smaller MPC only remained unchanged. As we concluded, combination RM/DC presents synergistic effect to MRSA 01 and indifference effect to MRSA 03. Therefore, these above indicated that the agent with a larger MPC has greater mutation-preventing potency than the gent with a smaller MPC in a drug combination, whatever the combination shows synergy or not, and probably whatever the drug concentration is less than their MICs, from their MICs to MPCs, or more than their MPCs since the concentrations of DC ranged from less than its MICs to MPC (those of RM less than its MICs). These results are in accordance with the regularity conclusion (11) in Table 2 of Section 2.1 and that drawn from Section 2.2. These results indicated that it preferentially occurs for the agent with the smaller MPC if the resistance is unavoidable for a drug combination, depending on whether its MSW was closed at the combinational ratio of two agents. Conversely, it preferentially occurs for the agent with larger MPC if the susceptibility is be enhanced for a drug combination, similarly depending on whether its MSW was closed at the combinational ratio of two agents. 

It is worth noting that the combination RM/DC showing indifference against MRSA 03 has greater mutation-preventing potency than that showing synergism against MRSA 01, and even the susceptibility of RM with the larger MPC to MRSA 03 is enhanced while that of DC with the larger MPC to MRSA 01 remains unchanged. According to the hypotheses of MSW and MPC, maintaining drug concentrations above MPC throughout therapy can prevent resistance. It was found that all the concentrations of two agents in combination RM/DC against MRSA 01 were less than their MICs alone. However, all the concentrations of DC in combination RM/DC against MRSA 03 were greater than its MPC (0.39 μg/mL) alone except for those in combinational proportions 8:1 (0.33 μg/mL) and 4:1 (0.34 μg/mL) in which the concentrations of DC are more than its MIC and very closed to its MPC, even probably more than the actual MPC since the MPC was determined by plate method with linear concentration decrease 20% (from 0.39 μg/mL to the next concentration of 0.312 μg/mL). These above findingfs further confirmed that the actual effects of drug combinations preventing AMR are not only related to the combinational proportions of two agents but also to the applied concentrations of two agents and depend on whether the concentration of any one of two agents in a drug combination is larger than itself MPC. Namely, the resistance would likely occur when both concentrations of two agents in a combination are lower than themselves MPCs, and while the susceptibility remains unchanged or is enhanced when the concentration of any agent in a combination is larger than itself MPCs. These completely coincided with the hypotheses of MSW and MPC for drugs applied in alone [17].

#### 2.4.2. Susceptibility Changes of MRSA to Antibiotics after Exposed to Designed Combinations

According to the hypotheses of MSW and MPC, maintaining drug concentrations above MPC throughout therapy can severely restrict the acquisition of drug resistance, while the resistance will be easy to acquire when the concentrations of antimicrobial agents fall into the range of MIC to MPC. From Table 6, it was indicated that combination RM/DC can prevent resistance from MRSA 03 when the concentration of DC with smaller MPC in the drug combination increases to above its MPC, along with the concentration of RM less than MIC. Whether this mutation-preventing effect can be obtained by increasing the concentration of RM with larger MPC in the drug combination increases up to above its MPC, and whether similar effect for MRSA 01 can be obtained by increasing the concentration of RM with smaller MPC in the drug combination up to above its MPC. To further explore these and improve those conclusions in Section 2.4.1, many combinational proportions from 3200:1 to 100:1 for MRSA 03, and from 16:1 to 1:32 for MRSA 01 were designed, together with various concentrations less than MIC, from MIC to MPC, or more than MPC, of two agents (Table 7).

As we observed and analyzed above, the resistance likely occurred when both concentrations of two agents were lower than themselves MPCs. From Table 7, the resistance occurred for MRSA 01 to combinational proportions 8 to 10, and MRSA 03 to those 8 to 11 and 20 to 25. In accordance with that no resistance occurred for MRSA 01 to the agent with its SI less than 1 in proportions 1 to 7 (Table 6), there was no resistance for MRSA 01 to DC with its SI less than 1 in proportions 8 to 10 (Table 7). Simultaneously, there was no resistance occurred for MRSA 03 to DC which all the SIs are less than those of RM for proportions 8 to 11, and 20 to 25 (Table 7). Therefore, we deduced that no resistance occurs for the agent with smaller SI in combination if the resistance is unavoidable for pathogenic bacteria to a drug combination (namely, the collateral sensitivity occurred), and this also coincided with ascertained conclusion that the agent with larger MPC in a drug combination has greater potency preventing the AMR. Here, the SIs of two agents in a combination can be calculated from established equations *y* = a_1_*x*^b1^ and *y* = a_1_*x*^b1^, or roughly obtained from the curve outliers like Figure 1. 

Simultaneously, it was also found that good mutation-preventing effects can be obtained when the concentration of the larger proportional agent in an antimicrobial combination is more than or equal to its MPC alone, such as the RM concentrations of 10.24 and 256 μg/mL, respectively, for proportions 11 to 13 against MRSA 01 and proportions 14 to 19 against MRSA 03, and the DC concentrations from 2.56 to 10.24 μg/mL for proportions 14 to 18 against MRSA 01, shown in Table 7. On exposure to these proportions of combination RM/DC, no resistance occurred for corresponding pathogenic bacteria to either agent. Otherwise, it is generally difficult to concurrently take into account for the mutation-preventing effects of both agents in antimicrobial combinations, such as MRSA 01 resistant to RM in proportions 8 to 10, and MRSA 03 resistant to RM in proportions 8 to 13, and 20 to 25. Combined with the results from Table 7, these concluded that increasing the concentration of the larger proportional agent in an antimicrobial combination up to more than or equal to its MPC alone can prevent the resistance of pathogenic bacteria to two agents and also bring out good mutation-preventing effects, enhancing the susceptibility of pathogenic bacteria to one of them, whatever the combination is synergistic or not. 

This was also supported by the results from published work [39] and the following facts. For example, after exposure to combinational proportions 11 to 18 in Table 7, MRSA 01 showed greater susceptibility to RM and DC. Otherwise, even in a synergistic combination, the inappropriate drug concentration would usually sacrifice the sensitivity of pathogenic bacteria to one agent in exchange for that to another. This was also confirmed by the fact that MRSA 01 and MRSA 03 remained sensitive to DC while simultaneously resistant to RM, from Table 6 and Table 7. Moreover, only the antimicrobial agent with the MSWs closed or with the smaller SI in a drug combination can probably prevent AMR, such as MRSA 01 to RM after exposed to proportions 5 or 7 in Table 6 or MRSA 03 to DC after exposure to proportions 8 to 13, and 20 to 25, when the concentrations of the agent with larger combinational proportion are less than itself MPC alone.

Another fact is that if it cannot be achieved for any agent to increase its concentration above its own MPC according to the actual situation, it is better to adjust the ratio of two agents to close their MSWs (such as proportions 5 and 7 against MRSA 01 in Table 6) or keep the SI of two agents enough difference (such as proportion 6 against MRSA 01 in Table 6, proportions 8 to 13, and 20 to 25 against MRSA 03 in Table 7), and lower the concentration of the aimed drug less than itself MIC for reducing the degree of drug resistance (compared proportions 8 to 13 with those 20 to 25 against MRSA 03 in Table 7) and the toxic and adverse effects [24]. Based on the above SI rule, we can select synergistic drug combinations as far as possible, adjusting the ratios of two agents for closing their MSWs, lowering their applied concentrations less than their MICs for acquiring better mutation-preventing effect, and reducing the toxic and adverse effects when good antibacterial effect is ensured. If the resistance is unavoidable, we can also adjust the ratios of two agents for changing their SIs to control the trajectories of collateral sensitivity, lowering the concentration of the agent with the SI more than 1 or the larger SI for reducing the degree of drug resistance and the toxic and adverse effects.

## 3. Discussion

In our previous work [24], we discovered that the mutation-preventing effect of a drug combination relates to the ratio of two agents and put forth some reasonable explanations for various and even contradictory reports involving antimicrobial combinations. Here, many correlations between various indexes, and regularity conclusions, laws and rules for drug combinations were further concluded from the mathematical and statistical analyses for our reported data. Based on all the above and combined with the verification and improvement from the antibiotic exposure experiments, some important principles were proposed for guiding the AMR-preventing practice, predicting the mutation-preventing effect, and controlling the trajectories of collateral sensitivity, and shown as following diagram. 

Based on our previous conclusion that the SI of one agent against a specific bacterial strain relates to the proportion of two agents in a drug combination [24], further analyses for those experimental data indicated that the correlation between the SI (*y*) of one agent and the ratio (*x*) of another to this agent in a drug combination presents the mathematical characteristics of power function *y* = a*x*^b^ (a>0), and some regularity conclusions and rules in Table 2 were discovered by communicating the mathematical characteristics of power functions for two agents with the related indexes of bacterial resistance in drug combinations, for roughly predicting the mutation-preventing effect and controlling the trajectories of collateral sensitivity. For a specific combination A and B (A/B), two equations *y* = a_1_*x*^b1^ (for A) and *y* = a_2_*x*^b2^ (for B) can be established from the calculation for two data pairs consisting of the SI of one agent (*x*) and the ratio (*y*) of another to this agent in a drug combination, and better from the curve fitting from seven to night data pairs such as the SI (*y*) at the ratio of another to the agent (*x*) 1/16, 1/8, 1/4, 1/2, 1, 2, 4, 8 or 16. Simultaneously, it is better for those data pairs to include (1, a). After both two equations were established for agents A and B, the SI of one agent at any ratio of two agents in a drug combination can be calculated for predicting and controlling the trajectories of collateral sensitivity when the applied concentrations of the agent with larger combinational proportion in a drug combination are less than its MPC. Another, three rules for two equations of two agents in a drug combination were obtained as (1) the curve must pass the dot (1, a); (2) a_1_ × MIC_1_ = a_2_ × MIC_2_; and (3) b_1_ + b_2_ = −1 for the self-checking of the equations and the qualitative judgment for the rationality of experimental data.

As the SI indicates the closed degree of the MSW and also reflects the mutation-preventing potency, we proposed the MPSI in a ratio range of two agents for an antimicrobial combination for evaluating the selectivity of mutation-preventing potency for two agents. Simultaneously, the calculation for this index was provided as Formula (2) in Section 4.1.2. Based on the analyses of our previous experimental data, we found that significantly positive correlation between the MPSI and the MPC or MIC ratio (MPC_larger_/MPC_smaller_ or MIC_larger_/MIC_smaller_) of two agents in a drug combination, especially between the MPSI and the MPC ratios. This indicated that the larger the MPC ratio (MPC_larger_/MPC_smaller_), the larger the MPSI (the larger SI_alone_/SI_combination_ divided by the smaller one), of two agents in drug combinations. Therefore, this conclusion is very helpful to predict and control the trajectories of collateral sensitivity (Figure 3). Inspired by the simultaneous correlations between the MPSI and the MPC, and the MPC and the MIC, the analyses for a correlation between the MIC and the MPC indicated that there is significantly positive correlation between the MIC and the corresponding MPC of an antimicrobial agent against the same pathogenic bacteria, instead of the low correlation between them reported in previous publications [17,24,31]. Therefore, the MPC of an antimicrobial agent can be roughly predicted from its MIC, and simultaneously the test procedure of actual MPC can be also simplified. More importantly, this would help to quickly and rationally select the appropriate drugs for a drug combination, without reference to their actual MPCs in some cases.

As we analyzed above, the inappropriate drug concentrations and combinational proportions would usually sacrifice the sensitivity of pathogenic bacteria to the agent with smaller MPC in exchange for that to another with larger MPC, even if synergistic combinations. This was confirmed by the fact that many resistances occurred for MRSA 01 and 03 to RM while the susceptibility of both two pathogenic isolates to DC increased, after exposed to various proportions and drug concentrations (Table 6 and Table 7). This phenomenon was namely collateral sensitivity (CS) [27], where resistance to one antimicrobial agent simultaneously increases the susceptibility to another. Along with the research on drug combination and cycling, etc. [48,49,50,51,52], CS-informed strategies were gradually developed, which would force bacteria to evolve resistance along a predictable trajectory, for preventing the AMR at the population level of bacterial communities ultimately and even reversing the resistance. However, it is still difficult to widely apply in clinic although many experiments were performed [48,49,50,51,52,53], since the simple and operable principles for guiding the practice of these strategies are rare [54]. Here, the positive correlations between the MPSI and the MPC ratio of two agents indicates that the MSW of the agent with larger MPC alone would be preferentially narrowed and even closed, and which will provide an important reference or guiding principle for predicting the bacterial responses to two agents and the evolutionary trajectories of AMR, during combinational therapy. Therefore, the positive correlation between the MPSI for drug combinations and the MPC ratio of two agents, provide a framework for rationally selecting drug combinations that limit resistance evolution.

As shown in Figure 3, to simultaneously prevent the resistance to two agents in combinations, the concentration of the agent with larger proportion should be applied larger than or equal to its MPC alone. Otherwise, the collateral sensitivity will probably occur. Usually, the MPCs of two agents used as drug combinations are different. Simultaneously, as above mentioned in Section 2.1 and Section 2.2, it is better for the MPCs of two agents to have enough difference for accurately predicting and controlling its trajectories if the collateral sensitivity occurs. Therefore, it is more reasonable and practicable for which drug in drug combinations to set as the larger proportional one to prevent the AMR, is the antimicrobial agent with larger MPC or that with smaller one? When the agent with higher MPC was set as a larger proportional one, increasing its concentration up to above the MPC would lead to a very larger concentration especially when the MPC difference of two agents in alone is very large (Table 7), and which likely lead to the possible toxic and side effects. Therefore, it is better to set the agent with smaller MPC as the larger proportional one in a drug combination and keep its concentration greater than or equal to its MPC alone. As above concluded in Section 2.1 and Section 2.2, the agent with larger MPC has greater mutation-preventing potency. Therefore, this will help the agent with larger MPC and smaller proportion to develop its mutation-preventing potency as far as possible. Of course, we can also increase the concentration of the agent with larger MPC up to greater than or equal to its MPC alone, to prevent the drug resistance when this agent has very low toxic and side effects. Since many natural products from plants including some herbs and Chinese traditional medicines [47,55,56,57,58,59], such as phenols [55], quinones, alkaloids [56], flavonoids [47,58], and terpenoids [59], generally have weaker antimicrobial activities and larger MPCs than antimicrobial agents and show good safety, it is encouraged to combine antimicrobial agents with them for preventing or delaying AMR [24,28].

Furthermore, the MSW of the agent with larger MPC can be preferentially narrowed, and even closed whatever the drug combination is synergistic or not. Also namely, the resistance to the agent with smaller MPC will be prior to occur, while the susceptibility to another with larger MPC will be preferentially enhanced. However, the actual effect of MSW closed depends on the proportion of an agent in drug combinations when the concentration of the agent with larger proportion is less than its MPC alone. Generally, the less the proportion for an agent in drug combinations, the more probable its MSW to be closed. Another, the less the FICI value, the more remarkable the synergistic effect of a drug combination, and the wider the proportional range of two agents closing each other’s MSW, as we concluded in our previous work [24]. Therefore, the actual effects on preventing the resistance to two agents in drug combinations are related to the ratio of two agents (more directly, their SIs) and the FICI value of the drug combination besides some principles shown in Figure 3, when the concentration of the agent with larger proportion is less than its MPC alone. 

As shown in Figure 3, the ratio and the applied concentration of two agents, and the FICI of the combination should be considered as three key factors of drug combinations preventing the resistance and predicting the trajectories of collateral sensitivity. Simultaneously, C_A_ ≥ MPC_A_ can be transformed as C_A_/MPC_A_ ≥ 1, and C_A_ < MPC_A_ can be transformed as C_A_/MPC < 1. According to Figure 3, this means that C/MPC ≥ 1 for the agent with larger proportion in a drug combination would prevent the bacterial resistance occurring to both two agents, and the larger the C/MPC value of this agent, the better the potency preventing the AMR. However, the C/MPC < 1 for the agent with larger proportion led to collateral sensitivity, and the smaller the 1/MPC of an agent in drug combinations, the greater its potency for preventing resistance to itself. These indicated that the C/MPC for the agent with larger proportion in drug combinations is a key for judging whether the resistance and the collateral sensitivity occur to two agents. This was indirectly and partly supported by many previous papers [19,21,42,60,61]. Simultaneously, the MPC is a specific index of an antimicrobial agent, relating the resistance of a certain pathogenic isolate, according to the hypotheses of MSW and MPC. Therefore, the reciprocal of MPC (1/MPC) can be considered as a stress factor for an antimicrobial agent to pathogenic bacteria, according to the above response of bacteria to antimicrobial agents, such as the susceptibility of pathogenic bacteria to the drug with larger MPC in a combination preferentially remains unchanged, and even sometimes is enhanced. The smaller the MPC, the larger the stress, and the easier the resistance is to occur. Conversely, the larger the MPC, the smaller the stress, and the more difficult the resistance is to occur. 

After in vivo administrated with antimicrobial combinations, the concentrations of two agents will drop below their MPCs sooner or later, and simultaneously the ratios of two agents in various tissues will also change as both agents have different pharmacokinetic characters. These will increase the complexity of the drug combinations that prevent resistance and fluctuate or even invert the anticipated effects [13,24,62]. Therefore, a specific analysis should be considered according to the practical application although the above regularity conclusions and laws had been drawn. For examples, (1) two agents with similar pharmacokinetic characters should be encouraged to be selected as far as possible for antibacterial combinations [17,24]. (2) The more significant the synergistic effect of two agents in a drug combination, the wider the proportional range of two agents closing each other’s MSW, and the more favorable to prevent resistance (Figure 3) [24,63]. Therefore, two agents with the FICI value as small as possible should be selected for drug combinations. Furthermore, we can discover synergistic combinations as quickly as possible according to the conclusion that antimicrobial agents targeting identical macromolecular biosynthesis pathway while different action sites (mechanisms) have a great potency to discover synergistic combinations [10,24]. (3) It is encouraged to set the agent with smaller SI as the larger proportional one, since the concentrations of the agent are unavoidable to fall into its MSW when its applied concentrations in human body are more than its MPC. (4) For the same pathogenic bacterium, the more susceptible to two agents, the larger the probability discovering synergistic combinations [12,24]. Therefore, it should be encouraged to select two agents which pathogenic bacteria are susceptible to for drug combinations, as proposed in our previous paper [24]. Moreover, a new antimicrobial agent in combination with another synergistic one, as a regular combination or even a hybrid antibiotic like rifamycin-quinolone [64], should be encouraged to be approved [24], since the resistance to new antimicrobial agents applied in alone would be emerged soon after they are approved for marketing.

As metabolized by human body, the concentrations above the MPC of the agent with larger proportion will unavoidably drop below its MPC, and the collateral sensitivity would be inevitably occurred. Therefore, it is difficult to simultaneously prevent the resistance of pathogenic to two agents in practice. To keep the desired mutation-preventing effects and force bacteria to evolve the resistance along a predictable trajectory, theoretically we can make the ratio of two agents fluctuate in a narrow range, which the mutation-preventing effect or the resistance trajectory remains unchanged, by selecting two agents with similar pharmacokinetic characters for a drug combination, such as a range from 1:4 to 1:8 for RM against MRSA 03, and from 8:1 to 1:8 for DC against MRSA 01. Moreover, according to the MSW closed trend deduced from the monotonically decreasing property of power function *y* = a*x*^b^ (generally, b < 0), it can deduce that the wider the range of the MSWs closed, the less the similarity requirements, keeping desired mutation-preventing effects or identical resistance trajectory, for the pharmacokinetic characters of two agents. On the other hand, the larger the similarity for the pharmacokinetic characters of two agents, the narrower the range required for the MSWs closed of two agents. Combined with the tendency correlation between the MSWs closed and the FICI of two agents in drug combinations [24], it showed that the more the synergism, the less the similarity required for the pharmacokinetic characters, of two agents to prevent the AMR. Simultaneously, the larger the similarity for the pharmacokinetic characters, the less the synergism required for two agents. Therefore, we may select two antimicrobial agents with synergistic effect and similar pharmacokinetic as far as possible for drug combinations, for simultaneously preventing the resistance of pathogenic to two agents, avoiding the collateral sensitivity, or predicting and controlling the resistance trajectory.

As above concluded, the C/MPC for the agent with larger proportion in drug combinations is a key for judging whether the resistance occurs and predicting the mutation-preventing effects. Therefore, we can keep the steady-state plasma concentration (*C*_SS_) of the larger proportional agent more than its MPC, by multiple administrations with proper adjustment to dose and interval time, for simultaneously preventing the resistance of pathogenic to two agents. Simultaneously, some important parameters AUC_24_/MPC, C_max_/MPC, and *f*%T > MPC for the larger proportional agent can be also considered as the explorable factors for preventing or delay the resistance, referring previous proposals [19,21,42,63]. On the other hand, this maybe a reason that some contradictory results could be occasionally drawn from different experiments for drug combinations, when only parameters AUC_24_/MPC, C_max_/MPC, and *f*%T > MPC were explored without the consideration whether those parameters are used for the larger proportional agent or for the smaller one.

Another, the half time (*t*_1/2_) of two agents selected should not be too large for avoiding the concentrations of two agents staying in their individual MSWs for too long time, during the ascending and descending phases of these two agents. If it is unable to simultaneously prevent the resistance to two agents, we can keep the proportion of one agent larger and that of another smaller, for predicting and controlling the resistance trajectory according to the tendency correlation (Figure 1) between the MSWs closed and the ratios of two agents in drug combinations. Since *t*_1/2_ and clearance (*CL*) are two important factors reflecting the change of drug concentration along with the time, we can set a larger proportion for the agent with larger *t*_1/2_ and smaller *CL* in a combination. This would keep the concentration of the drug with larger concentration larger all the time and ensure the consistency for the trajectory direction of collateral sensitivity. Combined the conclusion that it is better to increase the proportion of the agent with smaller MPC to prevent the resistance or enhance the susceptibility to another, setting a larger proportion for the agent with smaller MPC, larger *t*_1/2_ and smaller *CL* in drug combinations should be encouraged. 

Furthermore, some drugs with weaker antimicrobial activity show lower selection stress, and have insufficient potency to screen the resistant isolate. Therefore, it is also encouraged to select one agent with weak antimicrobial activity for narrowing the MSWs of another with remarkable antimicrobial activity to prevent the resistance, by greatly increasing it proportion in the drug combination whatever synergistic one or not, while synergistic one is better. It is noteworthy that some weak antimicrobial agents have been widely applying in combination with other antimicrobial agents to obtain antimicrobial effect, such as clavulanic acid, sulbactam, trimethoprim, and sodium 4-amino salicylate (with other anti-tuberculosis agents). However, related research had been rarely performed for them on the effect preventing the AMR except sulbactam combined with tigecycline [11]. Therefore, it is better for them to be reconsidered the rationality in preventing the AMR. In fact, the rationality of the classic antibacterial combination sulfamethoxazole/trimethoprim in the prevention of bacterial resistance is also worth exploring, since this drug combination cannot prevent the bacterial resistance although it has obtained good antibacterial effect. Another, many natural products from the plants have weaker antimicrobial activity than antimicrobial agents, such as phenols, quinones, alkaloids, flavonoids and terpenoids. Therefore, we can try to prevent or delay antimicrobial resistance by combining antimicrobial agents with one or more natural products from plants, herbs and Chinese traditional medicines. This has been also indicated from recent antimicrobial studies on plant natural products, such as proanthocyanidin [55], carnosic acid [59], and *α*-mangostin [65]. As plant natural products generally present weak antibacterial activity [47,58], they have large MPC according to the above positive correlation between the MIC and the MPC of an antimicrobial agent. Therefore, they probably present great mutation-preventing potency according to the conclusions in Section 2.1 and Section 2.2, and which may be the reason why it is difficult for pathogenic bacteria to be resistant to these compounds [55,56]. 

As above concluded, there is a positive correlation between the MPSI and the MPC ratio. It is worth noting that there is a vaguely correlation between the MPSI and the MPC when the MPC ratios of two drugs range from 0.24 to 4.2. This indicates that it is no obvious selectivity difference in pathogenic bacteria resistant to two agents in drug combinations when the MPC ratios of two drugs range from 0.24 to 4.2. To effectively predict and control the resistance trajectory, the difference between the MPCs of two drugs should be enough larger, at least larger than 4.2 times, or at least less than 0.24 times, and namely the MPC ratio of two drugs is larger than 4.2 or less than 0.24. Otherwise, which one in drug combinations bacteria preferentially resistant to probably depends on the most labor-saving rule of life, and which would possibly relate to the antimicrobial mechanisms of two drugs. 

Although only gram-positive bacteria were used for the test experiments, there is enough reason to infer that these above correlations, conclusions and laws are also applicable for gram-negative bacteria, and which was also confirmed by the fact that many related results from similar experiments on drugs against gram-negative bacteria coincide with them [10,34,35,40,54,55]. 

All above together, a preliminary scheme for antimicrobial combinations to prevent AMR was proposed as a foundation for subsequent improvement and clinic popularization and shown as Figure 4.

These above conclusions were drawn from double drug combinations. It is reasonable to infer that most of them are also applicable for triple or multiple drug combinations (namely, tri-drug or multi-drug combinations) since there are reasons to believe that bacteria should present similar response to antimicrobial agents. Therefore, some similar conclusions should be probably deduced and discussed although they need to be further verified, as follows:

(1) The C/MPC for the agent with larger proportion in drug combinations is also a key for judging whether the resistance and the collateral sensitivity occur to two agents. This means that the C/MPC ≥ 1 for the agent with largest proportion would prevent the bacterial resistance to all agents in the drug combination, and the larger the C/MPC value of this agent, the better the potency preventing the AMR. However, the C/MPC < 1 for the agent with largest proportion would lead to the collateral sensitivity, and the smaller the 1/MPC value of an agent in drug combinations, the larger its potency preventing the resistance to itself. 

(2) Which drug the susceptibility of pathogenic bacteria to remains preferentially unchanged and even is enhanced in multiple drug combinations can be judged by the following function.
(3)$$\{\begin{array}{c} \max\left\{MPC_{1},\;MPC_{2},MPC_{3}\;,\ldots ,\;MPC_{n}\right\}\\ s.t.\;n=2,\;3,\;4,\;5,\ldots \ldots \; \end{array}$$
where *n* is the number of compounds constituting a drug combination, and it is better for the *MPC* ratio of the first drug to the second one to be larger than 4.2, when their MPCs are sorted from the large to the small. 

It is worth noting that this is equivalent to antimicrobial agents applied in alone when *n* is equal to 1. At this moment, the above conclusions completely coincide with the hypotheses of MSW and MPC. Similar to double drug combinations, the smaller the proportion of the drug with largest MPC is, the more preferential the susceptibility of pathogenic bacteria to it remains unchanged or even enhanced. Simultaneously, it is possible deduced that the combination can prevent all drugs to be resistant when the concentration of the largest proportional drug keeps above its MPC alone. Also, if the collateral sensitivity is unavoidable, which drug in the combinations the susceptibility of pathogenic bacteria to remained unchanged or enhanced directly depends on whether its MSW is preferentially closed, relating the MPC (or 1/MPC) of the agent in drug combinations.

(3) Possibly, the correlation between the SI (*y*) of one agent and the combinational ratio of others to it (*x*) for triple or multiple drug combination also presents similar characteristics of power function *y* = a*x*^b^ (a > 0), and possibly three rules for the regression equations of each drug in combinations can be also established as (1) the curve must pass the dot (1, a); (2) a_1_ × MIC_1_ = a_2_ × MIC_2_ = a_3_ × MIC_3_ = … … = a_n_ × MIC_n_; and (3) b_1_ + b_2_ + b_3_ + …… + b_n_ = −1 or 1–n.

From the in-depth analyses for the experimental data reported by us and the antibiotic exposure experiments to drug combinations with different drug concentrations and various proportions, together with above discussions, some important discoveries, correlations and laws, principles, proposals, and hypotheses related to the prevention of AMR were summarized as follows:

(1) The correlation between the SI (*y*) of one agent and the ratio (*x*) of another to it for drug combinations presents the characteristics of power function *y* = a*x*^b^ (a > 0), and three rules for the equations of two agents were concluded as (1) the curve must pass the dot (1, a); (2) a_1_ × MIC_1_ = a_2_ × MIC_2_; and (3) b_1_ + b_2_ = −1. Based on this correlation, the SIs of one agent at any ratios of two agents can be calculated for predicting the resistance and controlling the resistance trajectory.

(2) A new concept of MPSI and its calculation formula were proposed for evaluating the mutant-preventing potency, while the actual effects depend on whether the MSW of drugs closed and its degree, relating to the concentrations and ratios of two agents, and the FICI value of the combination.

(3) The positive correlation between the MPSI and the MPC (or MIC) ratio of two agents was established, especially when the MPC (or MIC) ratios of two drugs are larger than 4.2 (or 1.5) or less than 0.24 (or 0.66). From this, we can simply predict and control the resistance trajectory using the MPC (or MIC) ratio of two agents instead of MPSI.

(4) The larger (more than 4.2) or smaller (less than 0.24) the MPC ratio of two drugs in a drug combination is, the more probable the sensitivity of pathogenic bacteria to the drug with larger MPC remains unchanged or is enhanced. Therefore, enough difference between the MPCs of two agents in a drug combination will be helpful to predict and control the resistance trajectory if the collateral sensitivity is inevitable.

(5) The C/MPC of the agent with larger proportion is a key for drug combinations to judge whether the resistance and the collateral sensitivity occur to two agents. Simultaneously, the reciprocal of MPC (1/MPC) alone was proposed as a stress factor for drug combinations to determine which one would present greater mutation-preventing potency and whether the susceptibility of pathogenic bacteria preferentially would be enhanced or remain unchanged to, predicting and controlling the trajectories of collateral sensitivity.

(6) Similar to double drug combinations, the C/MPC of the agent with larger proportion is also a key for tri-drug or multi-drug combinations to prevent the resistance and predict the trajectories of collateral sensitivity, and a function max{MPC_1_, MPC_2_, MPC_3_, …, MPC_n_} was similarly proposed for predicting and controlling the trajectories of collateral sensitivity. 

(7) Different from previous conclusions, there is a significantly positive correlation between the MIC and the MPC of an antimicrobial agent. Therefore, the MPC of an agent can be roughly calculated from its corresponding MIC.

(8) A diagram of the mutation-preventing effects and the resistant trajectories for drug combinations with different concentrations and proportional ratios of two agents was presented.

(9) A preliminary scheme for antimicrobial combinations preventing the AMR was also proposed. This includes the strategies and methods for preventing the resistant occurrence to two agents in drug combinations, and for the prediction and control of the resistance trajectory if the collateral sensitivity is unavoidable, and etc.

(10) To more effectively prevent the AMR, some principles of the drug selection for antibacterial combinations should be followed as far as possible, such as drugs which pathogenic bacteria are more susceptible to, drugs targeting identical macromolecular biosynthesis pathway while different action sites or mechanisms, drugs with similar pharmacokinetic character (such as similar absorption, distribution, *t*_1/2_ and clearance), combinational drugs presenting smaller FICI value (especially less than 0.5) as far as possible, enough difference between the MPC of two agents in drug combinations.

(11) Plant natural products with weak antimicrobial activity generally have far larger MPC than antimicrobial agents, and this may be the reason that it is difficult for pathogenic bacteria to be resistant to them. Simultaneously, it is very practical to combine them for enhancing the sensitivity of pathogenic bacteria to antimicrobial agents.

## 4. Materials and Methods

It had been concluded that the SI of one agent is closely related to the proportion of two agents in a drug combination in our previous work [24]. However, no specific mathematical correlation was established, and it still unable to guide the practice of drug combinations preventing AMR. Here, the in-depth analyses for those data were performed for exploring various correlations and laws, and which were further verified and improved by the antibiotic exposure experiments. Based on these, possible predictors, laws, principles and schemes were proposed for guiding the AMR-preventing practice.

### 4.1. Analyses for the Data Reported by Us

According to the hypotheses of MSW and MPC, the smaller the SI of an antimicrobial agent is, the more difficult the resistant occurrence is [17]. So, those experimental data in Table 1 and Table 2 reported by us [24] were further analyzed for discovering more information, possible correlations, and laws for preventing AMR. These data included the MICs and MPCs of five antimicrobial agents roxithromycin (RM), doxycycline (DC), vancomycin (VM), ofloxacin (OX) and fosfomycin (FF) in alone against three MRSA isolates, together with their MPCs in combinations RM/DC, VM/OX and VM/FF respectively with seven proportions against those three isolates.

#### 4.1.1. Correlation between the SI of One Agent and the Ratio of Two Agents in a Combination

The SIs in alone (MPC_alone_/MIC_alone_) and combination (MPC_combination_/MIC_alone_) were respectively calculated according to the hypotheses of MSW and MPC. Next, the correlation between the SI (*y*) of one agent in a drug combination and the ratio (*x*) of two agents was respectively analyzed for two agents using Microsoft Excel software, and which presents possible regression equations (Table 1), respectively together with their correlation coefficients (*r*) and their coefficients of determination (*R*^2^), for various proportions of two agents in a drug combination against a specific pathogenic bacterium. Using *r*-test, the statistical significances (*α* set as 0.05) were analyzed for these regression equations. Depending on the *R*^2^ value, the goodness of fit was compared for the regression equations. After the similarity characteristics were analyzed for these regression equations, probably functions were established for showing the correlation between the SI (*y*) of one agent in drug combinations and the ratio (*x*) of another to this agent. 

Further, more information, correlations, and laws preventing AMR from established functions of drug combinations were explored. Using mathematical deduction, the communications between the mathematical characteristics of the functions and the indexes related bacterial resistance were calculated for laws (1) to (5) in Table 2. Observed from those regression equations in Table 1, rules (6) and (10) in Table 2 were deduced, and further confirmed using mathematical statistics methods. For two functions *y* = a_1_*x*^b^_1_ and *y* = a_2_*x*^b^_2_ presented by two agents (their MICs and MPCs alone respectively marked as MIC_1_ and MIC_2_, and MPC_1_ and MPC_2_) in a drug combination, the correlations between the ratio of a_1_/a_2_ (*x*) and the MIC_2_/MIC_1_ (*y*) for law (7) in Table 1, and the ratio of b_1_/b_2_ (*x*) and the MPC_2_/MPC_1_ (*y*) for law (12) in Table 1 were established similarly using Microsoft Excel software. Laws (8) and (11) were observed from those regression equations in Table 1, and Laws (9) and (13) were respectively deduced from laws (6) and (7), and (11) and (12). According to similar analyses to function *y* = a*x*^b^, the communications between the mathematical characteristics of the function *y* = aln(*x*) + b and the indexes related bacterial resistance were performed. Another, based on these correlations and laws from function (1) *y* = a*x*^b^ in Table 2, some typical curve outlines, showing the representative correlations between the SI of one agent and the ratio of two agents in drug combinations were drawn, and the detail procedure was also presented.

#### 4.1.2. MPSI and Correlations between the MPSI and the MIC, MPC or SI Ratio of Two Antimicrobial Agents in a Drug Combination

For a drug combination, the MSW-closed degree of one agent in a ratio range of both two agents can reflect its ability to prevent bacterial resistance. As we analyzed above, inappropriate drug concentrations and combinational proportions usually sacrifice the sensitivity of pathogenic bacteria to one agent in exchange for that to another one, namely, collateral sensitivity occurs. To predict and control the trajectory of collateral sensitivity when two agents form a combination, the ratio for the maximum potency narrowing the MSW of two agents, in a ratio range (such as 1:16 to 16:1, or 1:64 to 64:1) of a drug combination, defined as mutation-preventing selection index (abbreviated as MPSI), of two agents in a ratio range (such as 1:16 to 16:1, or 1:64 to 64:1) of a drug combination, can be respectively calculated using Formula (2), for showing the potency difference preventing AMR.
(2)$$\begin{array}{cc} \mathrm{MPSI} & =\frac{{\mathrm{The}\;\mathrm{maximum}\;\mathrm{potency}\;\mathrm{for}\;\mathrm{narrowing}\;\mathrm{the}\;\mathrm{MSW}}_{A}}{{\mathrm{The}\;\mathrm{maximum}\;\mathrm{potency}\;\mathrm{for}\;\mathrm{narrowing}\;\mathrm{the}\;\mathrm{MSW}}_{B}}\\ & =\frac{\frac{{\mathrm{SI}}_{A}\;\mathrm{alone}}{{\mathrm{minimum}\;\mathrm{SI}}_{A}\;\mathrm{in}\;\mathrm{all}\;\mathrm{set}\;\mathrm{proportions}\;\mathrm{of}\;a\;\mathrm{combination}}\;}{\frac{{\mathrm{SI}}_{B}\;\mathrm{alone}}{{\mathrm{minimum}\;\mathrm{SI}}_{B}\;\mathrm{in}\;\mathrm{all}\;\mathrm{set}\;\mathrm{proportions}\;\mathrm{of}\;a\;\mathrm{combination}}}\\ & =\frac{\frac{{\mathrm{MPC}}_{A}\;\mathrm{alone}}{{\mathrm{The}\;\mathrm{minimum}\;\mathrm{MPC}}_{A}\;\mathrm{in}\;\mathrm{all}\;\mathrm{set}\;\mathrm{proprtions}\;\mathrm{of}\;a\;\mathrm{combination}}}{\frac{{\mathrm{MPC}}_{B}\;\mathrm{alone}}{{\mathrm{The}\;\mathrm{minimum}\;\mathrm{MPC}}_{B}\;\mathrm{in}\;\mathrm{all}\;\mathrm{set}\;\mathrm{proprtions}\;\mathrm{of}\;a\;\mathrm{combination}}} \end{array}$$
where, A and B were two antimicrobial agents in drug combinations; MSW_A_ and MSW_B_ were the MSWs of A and B; SI_A_ and SI_B_ were the SIs of A and B; and MPC_A_ and MPC_B_ were the MPCs of A and B. 

As we explained in paragraph 4 of Section 2.2., the more the MPSI value deviating 1.0, the larger the potency difference narrowing the MSW of two agents A and B in drug combinations. However, how much the MSW_A_ or MSW_B_ can be narrowed depending on the ratio of two agents in a drug combination [24], and the resistance to the agent with smaller potency narrowing the MSW would preferentially occur when collateral sensitivity is unavoidable. As the determination for MPSI is relatively complicated, whether there are simpler parameters replacing MPSI to approximately predict the trajectory of collateral sensitivity. Therefore, the correlations between the MIC, MPC or SI ratios in alone and the MPSIs (namely, the tested MPSI in Appendix A), of two antimicrobial agents in drug combinations, were analyzed based on those data in Table 1 and Table 2 previously reported by us [24], and the corresponding regression equations were also established, using Microsoft Excel software. When the MPSI is less than 1, the reciprocals of the MPSIs were taken, and correspondingly the reciprocals of the MIC, MPC, or SI ratios were also taken for the analysis of the correlation between the MIC, MPC, or SI ratio and the MPSI.

Another, the tested MPSI (1:1) was calculated from the experimental data in Table 1 and Table 2 previously reported by us [24], and simultaneously the calculated MPSIs and the calculated MPSIs (1:1) had been respectively calculated from two equations *y* = a_1_*x*^b1^ and *y* = a_2_*x*^b2^ (or *y* = a_1_ln(*x*) + b_1_ and *y* = a_2_ln(*x*) + b_2_) for two agents in a drug combination, according to Formula (2). To explore whether the tested MPSI can be replaced by the tested or calculated MPSI (1:1) for evaluating the potency difference preventing AMR of two agents in a drug combination, the correlations between the tested MPSI and the tested MPSI (1:1), and the tested MPSI (1:1) and the calculated MPSI (1:1) were further analyzed using Microsoft Excel software. 

#### 4.1.3. Correlation between the MPC and the MIC of an Antimicrobial Agent

Same to our previous reported [24], many papers [17,29,30,31] concluded that there was low correlation between the MIC and the MPC of an antimicrobial agent, and the MPC couldn’t be predicted from the MIC. Inspired by the correlation between MPSI and MPC (or MIC), the correlation between the MIC and the MPC was reanalyzed based on one hundred and eighty-one of data pairs reported in fourteen papers [11,12,23,24,32,33,34,35,36,37,38,39,40,41], using Microsoft Excel software. These data pairs include the MIC and the corresponding MPC, of various antimicrobial agents with different classes against representative pathogenic bacteria and shown in Appendix A. To obtain more intuitive visual effects, the correlation analyses were further performed using Microsoft Excel software after these data pairs (MIC, MPC) were respectively transformed into the natural logarithm (log_10_) of the MIC and the MPC. After this, using another forty-six data pairs (MIC, MPC) (Appendix A) reported from other papers [42,43,44,45,46], this correlation was further verified by comparing the calculated MPC with the tested one.

#### 4.1.4. Statistical Analysis

All regression equations, together with their correlation coefficients (*r*) or and their coefficients of determination (*R*^2^), were calculated using scatter plot and curve fitting tools in Microsoft Excel software. Using statistical *r*-test, the significances for the correlations were calculated. The goodness of fit for the regression equations was measured from the comparison of their coefficients of determination (*R*^2^). The closer *R*^2^ is to 1, the better the fit, and the closer the predicted value is to the actual one as a whole.

### 4.2. Verification and Improvement for Regularity Conclusion by Antibiotic Exposure Experiments 

Among the above three combinations, only RM/DC presented a synergistic antimicrobial effect [24]. It is interesting and fortunate that RM/DC not only presents synergistic inhibitory effect to MRSA 01 (the FICI ranged from 0.26 to 0.50) and 02 but also presents indifferent antimicrobial effect to MRSA 03 (the FICI ranged from 0.53 to 0.75). Therefore, the combination RM/DC was selected for the exposure experiments with MRSA 01 and 03 to verify and improve the above regularity conclusions and laws. In our previous work, the MICs (or MPCs) of RM to MRSA 01 and 03 were, respectively, 0.13 (or 0.32) and 32 (or 256) μg/mL, and those of DC to MRSA 01 and 03 were, respectively, 0.25 (2.56) and 0.13 (0.39) μg/mL. These data were also made sure again before this experiment.

#### 4.2.1. Antimicrobial Agents 

Roxithromycin (>940 U/mg) and doxycycline hydrochloride (88~94%) were obtained from BBI Life Sciences Corporation, Shanghai, China. Before use, roxithromycin (2.0 mg) was dissolved in 50 μL of dimethyl sulfoxide (DMSO), and then diluted with fresh sterile medium tested to obtain an initial concentration of 2.0 mg/mL. Correspondingly, 5% DMSO was prepared with fresh sterile medium tested as the blank control when need. The initial solution (2.0 mg/mL) of doxycycline was prepared by dissolving in sterile fresh medium tested. All the initial solutions were diluted with medium tested to obtain the desired concentrations.

#### 4.2.2. Isolates and Media

Two clinical MRSA isolates 01 and 03 were obtained from the Clinical Laboratory of the Second Affiliated Hospital, Sun Yat-sen University, and stored at −20 °C in 20% glycerol [66]. Mueller–Hinton broth (MHB) and Mueller-Hinton agar (MHA), purchased from Shanghai Sangon Bioengineering Co., Ltd., Shanghai, China, were used for all experiments except in vitro model test in which tryptic soy broth (TSB, Qingdao Haibo Biotechnology Co., Ltd., Qingdao, China) supplemented with 1% glucose was used. Prior to use, the isolate was cultured onto MHA plates at 37 °C, and pure colonies were cultured in 10 mL fresh MHB at 35 °C until the optical density (OD_600_) was approximately 0.60 to obtain bacterial suspension for test.

#### 4.2.3. Susceptibility Test

Following our previous method [24], all the MICs of the antimicrobial agents against MRSA isolates were calculated. Briefly, the tests were performed using broth microdilution method on 96-well plates in triplicate. After the plates were incubated at 37°C for 24 h, the MIC, defined as the lowest concentration of antimicrobial agent that completely inhibited bacterial growth in the micro-wells, was read by the unaided eye when the bacterial growth in blank wells was sufficient.

#### 4.2.4. Antibiotic Exposures

Various proportions of combination RM/DC, including different concentrations of two agents, were designed and shown in Table 6 and Table 7 for the antibiotic exposure experiments. In detail, the same seven proportions (**1** to **7**) of combination RM/DC as those in our previous work [24] were designed for exposure experiments of MRSA 01 and 03 to verify the regularity conclusions and laws. Based on the experimental results of MRSA 01 and 03 to these seven proportions, another eleven proportions (**8** to **18**) for MRSA 01 and eighteen proportions (**8** to **25**) for MRSA 03 were designed for further verifying and improving the regularity conclusions and laws.

Referring to previous methods [67,68], experiments of antibiotic exposure were performed. Briefly, bacterial suspensions with an OD_600_ of 0.2~0.3 were prepared from the purified cultures of MRSA isolates (OD_600_ ≈ 0.60) by the dilution with MHB medium. According to the designed proportions of combination RM/DC in Table 6 and Table 7, the antibiotic exposure to MRSA 01 or 03 was performed at 37 °C for 192 h in triplicate. During the exposure process, 0.5 mL of sample was taken every 24 h and centrifuged at 4000 rpm/min to remove the supernatant. The sediment was washed three times with normal saline and then diluted with MHB medium to obtain serial decimal dilutions (10^−1^ to 10^−7^). According to the drop plate method [69], five 5 μL drops from each dilution were placed onto a section of the MHA plate. Following incubation at 37 °C for 24 h, colonies from a sector from 10^−3^ to 10^−6^ were taken for the susceptibility test to RM and DC, and the MICs of the colonies, after consecutive passaging MRSA isolate 01 or 03 on antibiotic-free agar plates for six consecutive days, to RM and DC were further determined for the evaluation of the stability in drug resistance heritability. Finally, the susceptibility changes to RM and DC before and after the exposure experiments, were evaluated. 

## 5. Conclusions

In summary, applying various correlations, laws, and principles led to the conclusion that the ratio, concentration, and combinational FICI of two agents are three major factors determining mutation-preventing effects. More specifically, the C/MPC for the agent with the larger proportion in drug combinations can be considered a key in judging whether the resistance and the collateral sensitivity occur to two agents, and the reciprocals of MPC alone of two agents together with their SIs in combination are important factors to predict the mutation-preventing potency and control the trajectories of collateral sensitivity. From these, it is necessary to reevaluate the rationality and scientificity of many fixed-dose combinations being marketed in different countries and some related principles and guides for the clinical applications of antimicrobial combinations, for simultaneously obtaining good antimicrobial and AMR-preventing effects. Here, a diagram of the mutation-preventing effects and the resistant trajectories for drug combinations and a preliminary scheme for antimicrobial combinations preventing the AMR were proposed as a reference.

## Figures and Tables

**Figure 1 antibiotics-11-01279-f001:**
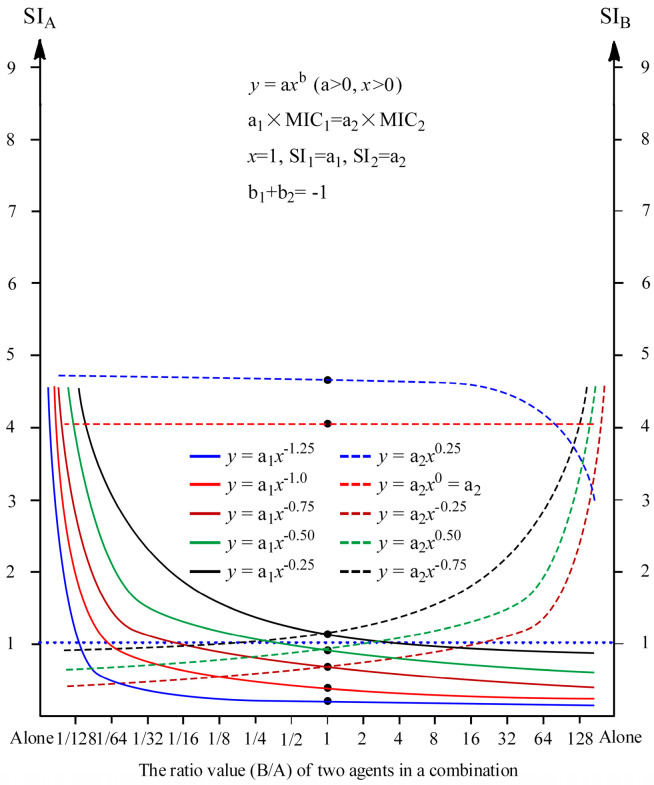
Diagrams of typical curve outlines showing the representative correlations between the SI (y) of one agent in drug combinations and the ratio (x) of two agents. Two functions *y* = a_1_*x*^b1^ and *y* = a_2_*x*^b2^, respectively passing the dots (1, a_1_) and (1, a_2_) when x = 1, were established for two agents A and B in a drug combination, and the longitudinal coordinates SI_A_ and SI_B_ were respectively the SIs of two agents A and B.

**Figure 2 antibiotics-11-01279-f002:**
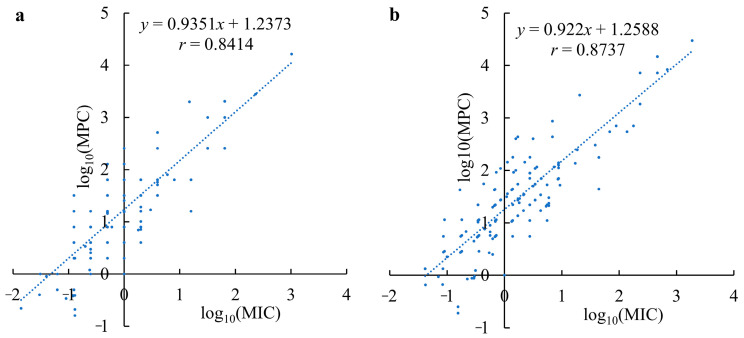
Correlations between the natural logarithm (log_10_) of MIC and that of MPC. (**a**), the concentration units of the MIC and MPC were μg/mL; (**b**), the concentration units of the MIC and MPC were μM/L.

**Figure 3 antibiotics-11-01279-f003:**
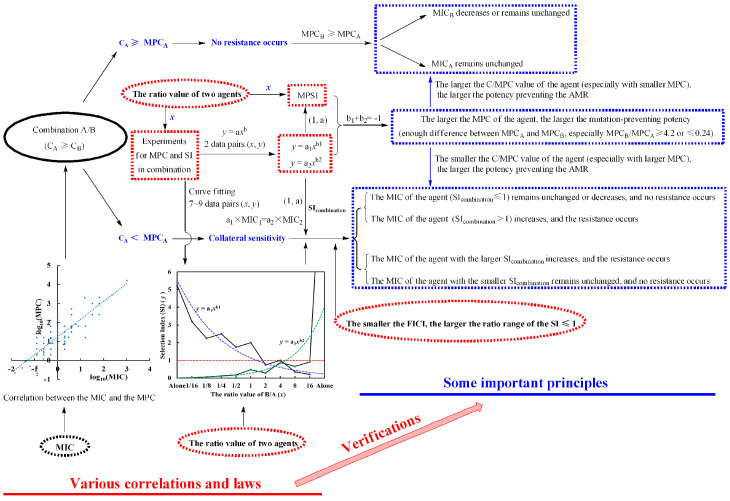
Diagram of the mutation-preventing effects and the resistant trajectories for drug combinations with different concentrations and ratios of two agents. Combination A/B consists of agents A and B, and C_A_ and C_B_ are the applied concentrations of agents A and B, respectively; MIC, minimal inhibitory concentration; MPC, mutant prevention concentration; FICI, fractional inhibitory concentration index; MPSI, mutation-preventing selection index; SI_combination_, the SI in combination of two agents; setting C_A_ larger than or equal to C_B_, and MPC_B_ larger than or equal to MPC_A_.

**Figure 4 antibiotics-11-01279-f004:**
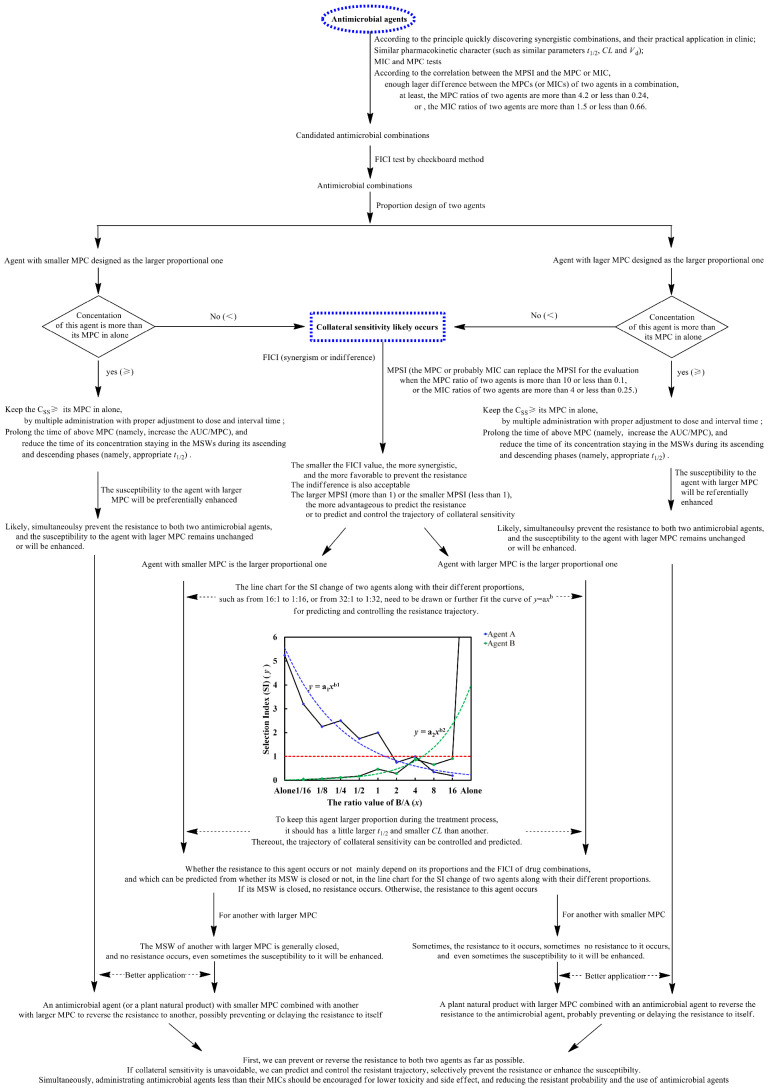
The preliminary scheme for drug combinations to prevent antimicrobial resistance.

**Table 1 antibiotics-11-01279-t001:** Correlation between the SI (y) of one agent in a drug combination and the ratio (x) of another to this agent. (n = 7).

Drug Combination ^a^	MRSA Isolates	Regression Equation ^b^		Correlation Coefficients (r) ^c^		Coefficient of Determination (R^2^)		Goodness of Fit ^d^	
		Type I	Type II	Type I	Type II	Type I	Type II	Type I	Type II
RM/DC	01	*y* = 0.3613*x*^−0.487^	*y* = −0.161ln(*x*) + 0.456	0.8292 *	0.9070	0.6876	0.8226	/	better
		*y* = 0.1838*x*^−0.618^	*y* = −0.126ln(*x*) + 0.2629	0.8472	0.8849	0.7177	0.7830	/	better
	02	*y* = 0.3392*x*^−0.318^	*y* = −0.108ln(*x*) + 0.3729	0.9804	0.9825	0.9611	0.9654	/	better
		*y* = 0.3669*x*^−0.678^	*y* = −0.289ln(*x*) + 0.5385	0.9773	0.9707	0.9552	0.9422	better	/
	03	*y* = 0.0124*x*^−0.925^	*y* = −0.018ln(*x*) + 0.0251	0.9982	0.8984	0.9964	0.8072	better	/
		*y* = 3.0580*x*^−0.071^	*y* = −0.203ln(*x*) + 3.0703	0.7740	0.7890	0.5990	0.6225	/	better
VM/OX	01	*y* = 2.6565*x*^−0.519^	*y* = −1.509ln(*x*) + 3.5271	0.7289 *	0.8002	0.5313	0.6404	—	better
		*y* = 2.6561*x*^−0.481^	*y* = −1.246ln(*x*) + 3.2771	0.9481	0.9608	0.8989	0.9232	/	better
	02	*y* = 1.7752*x*^−0.463^	*y* = −0.929ln(*x*) + 2.1907	0.9778	0.9565	0.9561	0.9149	better	/
		*y* = 1.7769*x*^−0.537^	*y* = −1.217ln(*x*) + 2.3864	0.9541	0.8703	0.9103	0.7575	better	/
	03	*y* = 3.6279*x*^−0.569^	*y* = −1.951ln(*x*) + 4.4196	0.9634	0.9132	0.9281	0.8340	better	/
		*y* = 0.7627*x*^−0.619^	*y* = −0.463ln(*x*) + 1.0457	0.9468	0.9794	0.8964	0.9592	/	better
VM/FF	01	*y* = 4.5965*x*^0.046^	*y* = 0.2592ln(*x*) + 4.8057	0.2128 *	0.2191 *	0.0453	0.0480	—	—
		*y* = 0.0716*x*^−1.049^	*y* = −0.122ln(*x*) + 0.1719	0.9634	0.9046	0.9281	0.8183	better	/
	02	*y* = 2.5407*x*^−0.278^	*y* = −0.747ln(*x*) + 2.8193	0.8573	0.8255	0.735	0.6814	better	/
		*y* = 0.1587*x*^−0.722^	*y* = −0.156ln(*x*) + 0.2629	0.8719	0.8680	0.7602	0.7535	better	/
	03	*y* = 10.24*x*^−0.129^	*y* = −1.464ln(*x*) + 10.541	0.7463 *	0.7187 *	0.5569	0.5166	—	—
		*y* = 0.1483*x*^−0.93^	*y* = −0.212ln(*x*) + 0.3019	0.9993	0.8900	0.9987	0.7921	better	/

^a^: RM: roxithromycin, DC: doxycycline, VM: vancomycin, FF: fosfomycin, OX: ofloxacin. ^b^: Type I and II were respectively fitted by power and logarithmic functions; for a specific MRSA isolate, the first equation presented the correlation between the SI (y) of RM (VM) and the ratio (x) of DC to RM (OX or FF to VM), and the second one presented that between the SI (y) of DC (OX or FF) and the ratio (x) of RM to DC (VM to OX or FF). ^c^: The significance level α was set as 0.05, and the critical value of r_0.975_ (5) is equal to 0.754; *: indicates no obvious correlation. ^d^: better, means the goodness of fit for the regression equation is better; /, indicates that is lower; —, shows no significance for the fit of the regression equation.

**Table 2 antibiotics-11-01279-t002:** Communications between the mathematical characteristics and the related indexes of bacterial resistance, of the functions for two agents in a drug combination **^a^**.

No.	*y* = a*x*^b^ (a > 0, *x* > 0)		*y* = aln(*x*) + b (b > 0, *x* > 0) ^b^	
1	Most, b < 0	Depended on the ratio of two agents in a drug combination, the MSWs of most agents can be generally closed whatever synergism or not.	Most, a < 0	Same
	Monotone decreasing function	(1) The SI of one agent in a drug combination decreases along with the proportional increase in another agent. Generally, the MSWs of one antimicrobial agent can be narrowed to some extent by in combination with another whether it is synergistic or not. (2) The larger the SI alone, the larger the |b|, and the faster the decrease in the SI.	Monotone decreasing function	Just replaces |b| with |a|.
	The curves must pass through the dot (1, a)	When the ratio of two agents is equal to 1 (*x* = 1), the SI is equal to a (*y* = a). (3) If a > 1, SI > 1, and the MSW is unclosed when *x* ≤ 1. (4) If a ≤ 1, SI ≤ 1, and the MSW is closed when *x* ≥ 1. (5) The smaller the value of a, the larger the ratio range of MSW closed, and the larger the probability of MSW closed.	The curves pass through the dot (1, b)	Just replaces a with b.
	(1) Two functions *y* = a_1_*x*^b^_1_ and *y* = a_2_*x*^b^_2_ were established for two agents in a drug combination, and their MICs and MPCs alone were repsectively marked as MIC_1_ and MIC_2_, and MPC_1_ and MPC_2_.	(6) a1/a2 = MIC2/MIC1 (namely a1 × MIC1 =a2 × MIC2)c in a drug combination. This rule was confirmed by the established correlation in Appendix A of Appendix A and indicates that one agent with larger MIC in a drug combination present a smaller a. Here, when *x* = 1, SI1/SI2 = a1/a2 = MIC2/MIC1.	Two functions *y* = a_1_ln(*x*)+b_1_ and *y* = a_2_ln(*x*) + b_2_ were established for two agents in a drug combination, and their MICs and MPCs alone were repsectively marked as MIC_1_ and MIC_2_, and MPC_1_ and MPC_2_.	(6) No relationship between b_1_/b_2_ = MIC_2_/MIC_1_ can be established, while one agent with larger MIC value also presents a smaller b value in a drug combination.
		(7) The larger the differeence between the MICs of two agents in a drug comination, the larger the differeence between the a values of both two equations. This was conclused from the positive correlation between the value of a_1_/a_2_ or a_2_/a_1_ (*x*) and that of MIC_2_/MIC_1_ or MIC_1_/MIC_2_ (*y*) ^c^ alone presents a linear equation *y* = 0.9602*x* or *y* = 0.9932*x*, respectively with the r value of 0.9993 or 0.9998, and the establishment of this correlation was detailed in Appendix A.		(7) Just replaces a with b, and a linear equation *y* = 1.9721*x* (*r* = 0.9910) was established and detailed in Appendix A.
		(8) The synergistic drug combination (with FICI ≤ 0.50) presents smaller value of a_1_ plus a_2_ (a_1_ + a_2_), and maybe both a_1_ and a_2_ ≤ 0.50 or (a_1_ + a_2_) ≤ 1. This was observed from Table 1, and need to be further verified by larger sample.		(8) Just replaces a with b, and maybe both b_1_ and b_2_ ≤ 1.0 or (a_1_ + a_2_) ≤ 2.
		(9) (6) and (7) indicate that the a value is related to the MIC of the agent and the FICI of a drug combination contained it. As a whole, the larger the MIC and the smaller the FICI, the smalller the a value, for an antimicrobial agent in drug combinations.		(9) Just replaces a with b.
		(10) b_1_ + b_2_ = −1. This rule was confirmed by the established correlation in Appendix A, which indicates that the decrease rate of SI for one agent slows down (namely, the |b| value decrease) when that for another speed up (namely, the |b| value increase), and even occasionally the SI for one agent increases along with the proportional increase of another agent (b > 0), in a drug combination.		(10) No relationship for a_1_ + a_2_ can be established.
		(11) As a whole, one agent with a larger MPC alone presents a larger |b| in a drug combination and show greater potency for narrowing the MSW and preventing the resistance.		(11) No correlation between the MPC and the |b| value was observed.
		(12) The larger the difference between the MPCs of two agents in a drug combination, the more obvious and larger the difference between the b values of both two equations. This was concluded from the correlation between the value (more than 1) of b_1_/b_2_ or b_2_/b_1_ (*x*) and that of MPC_1_/MPC_2_ or MPC_2_/MPC_1_ (*y*) ^d^ alone presents a linear equation *y* = 28.831*x* − 27.831, with the *r* value of 0.9985, and its establishment was detailed in Appendix A.		(12) Just replaces a with b, while the lower correlation between the value (more than 1) of a_1_/a_2_ or a_2_/a_1_ and that of MPC_1_/MPC_2_ or MPC_2_/MPC_1_ in alone^d^ presents a linear equation *y* = 29.956*x* − 28.956, with the *r* value of 0.9521, in Appendix A.
		(13) (11) and (12) indicate that the |b| value is largely related to the MPC alone of the agent in drug combinations.		
2	Occasionally, b > 0	Generally, the MSW cannot be closed except that a ≤ 1 (rarely), whatever synergism or not.	Occasionally, a > 0	Just replaces a with b

^a^: These communications were achieved by mathematical calculation and/or statistical processing after observed from those equations in Table 1. ^b^: The conclusions were described by comparison with function (1) *y* = a*x*^b^, according to similar analyses to function (1). ^c^: MIC, minimum inhibitory concentration, was calculated in mass concentration (μg/mL). ^d^: MPC was calculated in molar concentration (μM/L).

**Table 3 antibiotics-11-01279-t003:** The ratios for the MIC, MPC or SI alone of two antimicrobial agents in three drug combinations against three MRSA isolates, and the MPSIs of these combinations ^a^.

MRSA Isolates	Combinations ^b^ (A/B)	MIC Ratios ^c^		MPC Ratios		SI Ratios	MPSIs
		C_1_	C_2_	C_1_	C_2_	MPC/MIC	
01	RM/DC	0.520	0.276	0.082	0.044	0.158	0.082
	VM/OX	1.000	0.249	0.602	0.150	0.602	0.470
	VM/FF	0.031	0.003	0.015	0.001	0.493	0.002
02	RM/DC	1.000	0.531	0.080	0.042	0.080	0.040
	VM/OX	1.000	0.249	0.833	0.208	0.833	0.554
	VM/FF	0.063	0.006	0.016	0.002	0.256	0.008
03	RM/DC	246.154	130.678	656.410	348.475	2.667	3094.505
	VM/OX	0.250	0.062	1.875	0.468	7.500	0.778
	VM/FF	0.016	0.001	0.059	0.006	3.750	0.008

^a^: These data were calculated from the experimental data of Table 1 and Table 2 in our previous publication [24]. ^b^: RM, roxithromycin; DC, doxycycline; VM, vancomycin; FF, fosfomycin; OX, ofloxacin. ^c^: All ratios were calculated from agent A (the former) divided by agent B (the latter) in drug combinations, and the ratios for C_1_ and C_2_ were respectively calculated from the mass concentration (μg/mL) and the molar concentration (μM/L), of two antimicrobial agents.

**Table 4 antibiotics-11-01279-t004:** Regression equations between the MPSI (*y*) and the MIC (or MPC) ratio (*x*), of two agents in drug combinations. (*n* = 9) ^a^.

Independent Variable (*x*)	Regression Equation (Equation Number)	*r* ^b^	*R* ^2^	*r*_0.995_ (*n*-2)
MIC ratio	*y* = 12.2602*x* − 122.5719 (10.1 ≤ *x* ≤ 246.2) (1)	0.9719 **	0.9445	0.798
	*y =* 0.0450*x*^2^ + 1.4972*x* (1.5 ≤ *x* ≤ 246.2) (2)	0.9973 **	0.9945	0.798
MPC ratio	*y* = 4.7313*x* − 14.7324 (3.4 ≤ *x* ≤ 656.4) (3)	0.9973 **	0.9946	0.798
	*y =* 0.0009*x*^2^ + 4.1115*x* (4.2 ≤ *x* ≤ 656.4) (4)	0.9974 **	0.9948	0.798
	*y* = 4.7352*x* − 16.8261 (3.8 ≤ *x* ≤ 656.4) (5)	0.9973 **	0.9945	0.834
	*y =* 0.0009*x^2^* + 4.1116*x* (4.2 ≤ *x* ≤ 656.4) (6)	0.9973 **	0.9946	0.834

^a^: The mass concentration (μg/mL) of MIC and MPC are used for the analyses. Based on the data in Table 3, the reciprocals of the MPSIs were taken when the calculated value of the MPSI is less than 1, and correspondingly the reciprocals of the MIC, MPC, or SI ratios were also taken for the correlation analyses; the data pair (0.533, 1.285) was omitted when Equations (5) and (6) were established, and so eight data pairs were used for the establishment of Equations (5) and (6). ^b^: *r*, correlation coefficient; *r*_0.995_ (*7*) for Equations (1) to (4), and *r*_0.995_(*6*) for Equations (5) and (6) were the critical values when the significant levels *α* were set as 0.01; using *r*-test, the very significant difference (*P* < 0.01) was marked as **.

**Table 5 antibiotics-11-01279-t005:** Regression equations between the MIC (*x*) and the corresponding MPC (*y*), of antimicrobial agents against pathogenic bacteria (*n* = 181) ^a^.

Concentration Range	Regression Equation (Equation Number)	*r*	*r*_0.995_ (179)	*R^2^*
0.0312~1024 (μg/mL)	* y =* 16.025*x* (7)	0.980 **	0.20	0.9622
	* y =* 0.00006*x*^3^ − 0.0710*x*^2^ + 25.5154*x* (8)	0.9837 **		0.9677
	* y =* 0.9351*x* + 1.2373 ^b^ (9)	0.8414 **		0.7079
0.0414~1873 (μM/L)	* y* = 18.743*x* (10)	0.9354 **		0.8750
	* y* = 0.000004*x*^3^ − 0.0142*x*^2^ + 29.8848*x* (11)	0.9528 **		0.9079
	* y* = 0.922*x* + 1.2588 ^b^ (12)	0.8737 **		0.7633

^a^: *r*, correlation coefficient; *r*_0.995_ (179) was the critical values when the significant levels *α* were set as 0.01; using *r*-test, the very significant difference (*P* < 0.01) was marked as ******; *R*^2^ was coefficient of determination. ^b^: *x* and *y* were respectively taken from the natural logarithm (log_10_) of MIC and MPC for their more intuitive correlation.

**Table 6 antibiotics-11-01279-t006:** Susceptibility of MRSA isolates to antibiotics after exposed to combination roxithromycin/doxycycline (RM/DC) with seven proportions ^a^.

MRSA Isolates	FICI	Proportion Number	Ratio of Two Agents RM/DC	Antibiotic Concentration (μg/mL) RM/DC ^b^	Dose Range RM/DC (μg/mL)	Susceptibility Changes (×MIC) RM/DC ^c^	SI in Combination (RM/DC) ^d^
01	0.26~0.50	1	8:1	0.09/0.01	<MIC/<MIC	R (1024×)/―	>1/<1
		2	4:1	0.10/0.02	<MIC/<MIC	R (1024×)/―	>1/<1
		3	2:1	0.07/0.03	<MIC/<MIC	R (16~32×)/―	>1/<1
		4	1:1	0.08/0.08	<MIC/<MIC	R (32~64×)/―	>1/<1
		5	1:2	0.03/0.05	<MIC/<MIC	―/―	<1/<1
		6	1:4	0.04/0.15	<MIC/<MIC	R (256×)/―	1≈/<1
		7	1:8	0.01/0.12	<MIC/<MIC	―/―	<1/<1
03	0.53~0.75	1	8:1	2.64/0.33	<MIC/MIC~MPC	―/―	>1/>1
		2	4:1	1.34/0.34	<MIC/MIC~MPC	―/―	>1/>1
		3	2:1	0.86/0.43	<MIC/>MPC	―/―	>1/>1
		4	1:1	0.41/0.41	<MIC/>MPC	―/―	>1/>1
		5	1:2	0.20/0.41	<MIC/>MPC	―/―	<1/>1
		6	1:4	0.11/0.42	<MIC/>MPC	S (2×)/―	<1/>1
		7	1:8	0.07/0.56	<MIC/>MPC	S (4×)/―	<1/>1

^a^: FICI, fractional inhibitory concentration indexes; RM, roxithromycin; DC, doxycycline. ^b^: The concentrations of two agents in different proportions of combination RM/DC designed ac-cording to Table 2 in our previous paper [14] for the exposed experiment; ^c^: R, indicates that the MRSA isolate is resistant to the antibiotic; ―, indicates that the susceptibility of the MRSA isolate remains unchanged to the antibiotic; S, the susceptibility MRSA isolate to the antibiotic is enhanced. ^d^: The SIs of RM and DC for different proportions of combination RM/DC, ≤ 1, means the MSW was closed, and no resistance to the agent occurred according to the MSW hypothesis.

**Table 7 antibiotics-11-01279-t007:** Susceptibility changes of MRSA isolates to two agents after exposed to various proportions of combination roxithromycin/doxycycline (RM/DC) with different concentrations ^a^.

MRSA Isolates	FICI	Proportion Number	Ratio of Two Agents RM/DC	Antibiotic Concentration (μg/mL) RM/DC	Dose Range RM/DC (μg/mL)	Susceptibility Changes (×MIC) RM/DC ^b^	SI in Combination (RM/DC) ^c^
01	0.26~0.50	8	1:4	0.13/0.52	=MIC/MIC~MPC	R (4096×)/S (<1/4×)	1≈/<1
		9	1:8	0.13/1.04	=MIC/MIC~MPC	R (2×)/S (<1/4×)	<1/<1
		10	1:1	0.21/0.21	MIC~MPC/<MIC	R (4096×)/S (<1/4×)	>1/<1
		11	8:1	10.24/1.28	>MPC/MIC~MPC	S (<1/2×)/S (<1/4×)	>1/<1
		12	4:1	10.24/2.56	>MPC/=MPC	S (<1/2×)/S (<1/4×)	>1/<1
		13	16:1	10.24/0.64	>MPC/MIC~MPC	S (<1/2×)/S (<1/4×)	>1/<1
		14	1:16	0.16/2.56	MIC~MPC/=MPC	S (<1/2×)/S(<1/4×)	<1/NS
		15	1:16	0.32/5.12	=MPC/>MPC	S (<1/2×)/S (<1/4×)	<1/NS
		16	1:32	0.08/2.56	< MIC/=MPC	S (<1/2×)/S (<1/4×)	<1/NS
		17	1:32	0.16/5.12	MIC~MPC/>MPC	S (<1/2×)/S (1/2×)	<1/NS
		18	1:32	0.32/10.24	=MPC/>MPC	S (<1/2×)/S (<1/4×)	<1/NS
03	0.53~0.75	8	3200:1	128/0.04	MIC~MPC/<MIC MIC~MPC/<MIC	R (>8×)/―	SI_RM_ > SI_DC_
		9	1600:1	128/0.08		R (>8×)/(1/2~1×) ―	SI_RM_ > SI_DC_
		10	800:1	128/0.16	MIC~MPC/MIC~MPC	R (>8×)/―	SI_RM_ > SI_DC_
		11	400:1	128/0.32	MIC~MPC/MIC~MPC	R (>8×)/―	SI_RM_ > SI_DC_
		12	200:1	128/0.64	MIC~MPC/>MPC	R (>8×)/―	SI_RM_ > SI_DC_
		13	100:1	128/1.28	MIC~MPC/>MPC	R (4×)/―	SI_RM_ > SI_DC_
		14	3200:1	256/0.08	=MPC/<MIC	―(1~2×)/―	SI_RM_ > SI_DC_
		15	1600:1	256/0.16	=MPC/MIC~MPC	―(1/2~1×)/―	SI_RM_ > SI_DC_
		16	800:1	256/0.32	=MPC/MIC~MPC	―(1/2~1×)/―	SI_RM_ > SI_DC_
		17	400:1	256/0.64	=MPC/>MPC	―/―	SI_RM_ > SI_DC_
		18	200:1	256/1.28	=MPC/>MPC	―(1~2×)/―	SI_RM_ > SI_DC_
		19	100:1	256/2.56	=MPC/>MPC	―(1/2~1×)/―	SI_RM_ > SI_DC_
		20	3200:1	32/0.01	=MIC/<MIC	R (2×)/―	SI_RM_ > SI_DC_
		21	1600:1	32/0.02	=MIC/<MIC	R (2×)/―	SI_RM_ > SI_DC_
		22	800:1	32/0.04	=MIC/<MIC	R (2~4×)/―	SI_RM_ > SI_DC_
		23	400:1	32/0.08	=MIC/<MIC	R (4×)/―	SI_RM_ > SI_DC_
		24	200:1	32/0.16	=MIC/MIC~MPC	R (4×)/―	SI_RM_ > SI_DC_
		25	100:1	32/0.32	=MIC/MIC~MPC	R (2~4×)/―	SI_RM_ > SI_DC_

^a^: RM, roxithromycin; DC, doxycycline. ^b^: R, indicates that the MRSA isolate is resistant to the antibiotic; ―, indicates that the susceptibility of the MRSA isolate remains unchanged to the antibiotic; S, the susceptibility MRSA isolate to the antibiotic is enhanced. ^c^: SI_RM_>SI_DC_, judged from the monotonic decreasing characteristics of power functions *y* = a*x*^b^ (generally, b < 0); <1, ≈1, and >1 were calculated from the data in Table 2 of our previous work [24]; NS, not sure.

## Data Availability

The data presented in Table 1 and Table 2 in our previous paper [24], here reused as the raw data for the correlation analyses in Section 4.1, were reported by us at Springer Nature, and are available from the link at https://www.nature.com/articles/s41598-018-25714-z (accessed on 1 August 2022). All other data generated or analyzed during this study are included in this published article.

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
