# Peer review of "Drug Combinations to Prevent Antimicrobial Resistance: Various Correlations and Laws, and Their Verifications, Thus Proposing Some Principles and a Preliminary Scheme"

_antibiotics, 2022, doi:10.3390/antibiotics11101279_

Round 1
Reviewer 1 Report
In the Introduction section, I recommend highlighting the importance of the research topic and its applicability
Author Response
Point 1: In the Introduction section, I recommend highlighting the importance of the research topic and its applicability
Response: Thank you for your valuable suggestions!
According to your suggestions, we had already highlighted the importance of the research topic and its applicability as possible as we can, and improved the logic of the Introduction section. For example,
- In first paragraph, we inserted the following text: Thereby, it is desperate to discover some regularity conclusions and put forward some proposals and applicable schemes, for effectively guiding the practices of drug combinations preventing AMR, simultaneously avoiding the accelerated spread of AMR due to the abuse of drug combinations.
- The last paragraph of the Introduction section was rewritten as follows: Here, based on our previous data [24], the deeper analyses were performed for dis-covering probable correlations between the various indexes (such as MIC, MPC, fractional inhibitory concentration index (FICI) and mutant selection index (SI)), predictors, and laws, and then the analysis conclusions were verified and improved through the communications with the practical effects preventing AMR of different proportional combinations after antibiotic exposure experiments. Based on this, a preliminary scheme that can guide the practice of drug combination preventing AMR, together with some proposals and laws, was put forward for further experimental improvements and clinical trials.
- We had provided more detailed background in the second paragraph of revised manuscript. The content of the second paragraph was presented as follows: To prevent AMR, the hypotheses of mutant selection window (MSW) and mutant prevention concentration (MPC) were put forward by Zhao and Drlica [17]. According to these hypotheses, maintaining drug concentrations above its MPC throughout therapy can severely restrict the acquisition of drug resistance. Based on these hypotheses, many related experiments have been performed for discovering probable parameters to predict drug combination effects on the prevention of AMR or the selection of resistant mutant, and several MPC- or MSW-related parameters were proposed [18-22], such as MPC level (AAMPC), AUC24/MPC (Area under the concentration-time curve over 24 h divided by the MPC) and Cmax/MPC (Highest concentration divided by the MPC), and AUC24/MIC (AUC24 divided by the minimal inhibitory concentration) [23]. These results indicated that anyone of those predictors cannot be widely used to guide the practices of drug combinations preventing AMR.
These above revisions for your consideration, and thank you very much!
Some other revisions:
Besides the Introduction section, we had carefully performed extensive revisions throughout the manuscript including references, linguistic edit, expression, spelling, and grammar, etc. for your consideration.
Thank you very much!

Reviewer 2 Report
I enjoyed reading the article, I find it extremely interesting. Antimicrobial resistance is a serious threat to humans, combination therapy is proved to be economic indeed in fighting the resistance. Yes, the abuse of drug combinations can accelerate the spread of resistance. To guide the practice is essential.
Based on the previous work and research the diagram of the mutation-preventing effects and the resistant trajectories of drug combinations with different concentrations and ratio of two agents was presented, and the C/MPC for the agent with larger proportion in drug combinations may be the key to judge whether the resistance and the collateral sensitivity occur to two agents.
Author Response
Dear Reviewer,
My co-authors and I are very grateful to you for your valuable comments and suggestions. We have amended the manuscript according to the issues raised by you, and have pleasure to submit the revised version, together with the response to all points, for your consideration.
Many thanks for your kind attention!
Yours sincerely,
Ganjun Yuan
Here are our responses to your comments.
Point 1: I enjoyed reading the article, I find it extremely interesting. Antimicrobial resistance is a serious threat to humans, combination therapy is proved to be economic indeed in fighting the resistance. Yes, the abuse of drug combinations can accelerate the spread of resistance. To guide the practice is essential.
Based on the previous work and research the diagram of the mutation-preventing effects and the resistant trajectories of drug combinations with different concentrations and ratio of two agents was presented, and the C/MPC for the agent with larger proportion in drug combinations may be the key to judge whether the resistance and the collateral sensitivity occur to two agents.
Response: Thank you for your positive and good evaluation!
Other revisions:
Point 2: Are all the cited references relevant to the research? Can be improved
Response: Thank you for your valuable suggestion!
According to your suggestion, we had already checked the cited references, taking a new reference to replace reference 5 in the original manuscript, and inserting three closely relevant references 14 to 16 in the revised manuscript. To improve the relevance of the cited references to the research, we had also adjusted the cited location of some references for more appropriate citations instead of relatively vaguer ones in the original manuscript, especially in the Introduction section.
These revisions had been performed for your consideration.
Thank you very much!
Point 3: Are the methods adequately described? Can be improved
Response: Thank you for your valuable suggestion!
According to your suggestion, we had already detailed the methods for enhancing their repeatability, especially in sections “4.3.1” and “4.3.2”.
For example, in section “4.3.1”, we had inserted the sentences as follows: Using r-test, the statistical significances (α set as 0.05) were analyzed for these regression equations. Depending on the R2 value, the goodness of fit was compared for the regression equations.
In section “4.3.2”, we had inserted a paragraph as follows: Another, the tested MPSI (1:1) was calculated from the experimental data in Tables 1 and 2 previously reported by us [24], and simultaneously the calculated MPSIs and the calculated MPSIs (1:1) had been respectively calculated from two equations y = a1xb1 and y = a2xb2 (or y = a1ln(x) + b1 and y = a2ln(x) + b2) for two agents in a drug combination, according to formula (2). To explore whether the tested MPSI can be replaced by the tested or calculated MPSI (1:1) for evaluating the potency difference preventing AMR of two agents in a drug combination, the correlations between the tested MPSI and the tested MPSI (1:1), and the tested MPSI (1:1) and the calculated MPSI (1:1) were further analyzed using Microsoft Excel software.
Another, we had carefully performed extensive revision throughout the section “4. Materials and Methods”, including expression, spelling, and grammar. Simultaneously, we had checked the references.
These main revisions were performed for your consideration.
Thank you very much!

Reviewer 3 Report
The Manuscript named "Drug combinations to prevent antimicrobial resistance: some correlations, rules and laws, and a preliminary scheme" exerts a nice piece of the complex scientific art of work. I have a few concerns,
1. Line 287, before using the data from the previous publication did you ask for permission? or any ethical concern from the published journal!!
2. The statistical significance should be analyzed in another fitting and regression curve, as this whole paper stands on the statistical comparison mainly!
Author Response
Dear Reviewer,
My co-authors and I are very grateful to you for your valuable comments and suggestions. We have amended the manuscript according to the issues raised by you, and have pleasure to submit the revised version, together with the response to all points, for your consideration.
Many thanks for your kind attention!
Yours sincerely,
Ganjun Yuan
Here are our answers to your comments.
The Manuscript named "Drug combinations to prevent antimicrobial resistance: some correlations, rules and laws, and a preliminary scheme" exerts a nice piece of the complex scientific art of work. I have a few concerns,
Response: Thank you for your positive and good evaluation!
Point 1: Line 287, before using the data from the previous publication did you ask for permission? or any ethical concern from the published journal!!
Response: Thank you for your kind reminder!
According to your reminder, we had carefully checked whether it need to ask for permission or any ethical concern from the published journal, before using the data from our previous publication. The result indicate that this paper is an open access article distributed under the terms of the Creative Commons CC BY license, and we have the copyright and are not required to obtain permission to reuse this article.
The screen capture of the checked result was provided as follows for your consideration.
Thank you very much!
Point 2: The statistical significance should be analyzed in another fitting and regression curve, as this whole paper stands on the statistical comparison mainly!
Response: Thank you for your valuable suggestion!
You are right! Most conclusions of our manuscript mainly stand on the statistical comparison. When we were going to perform the statistical analyses for another fitting and regression curve you mentioned, we found we are not sure whether we had accurately understood another fitting and regression curve you mentioned.
So, we had tried our best to check and clarify the statistical analyses involved in our original manuscript one by one, and had performed some main revisions involved the following two sections for your consideration:
(1) For section “2.1 Correlation between the SI of one agent and the ratio value of two agents, in a drug combination”
Some expressions were vague for the conclusions from y = aln(x) + b (b > 0, x > 0) in the right column of Table 2, although we had simultaneously performed the statistical analyses in the original manuscript. Therefore, some revisions for the conclusions from y = aln(x) + b (b > 0, x > 0) in Table 2 were performed, and listed as follows:
- a) To clarify the method of the conclusions from y = aln(x) + b (b > 0, x > 0), we had revised the footnote b as “The conclusions were described by comparison with function (1) y = axb, according to similar analyses to function (1).”. Namely, the conclusions from y = aln(x) + b (b > 0, x > 0) were obtained using similar statistical analyses with those for y = axb.
- b) Simultaneously, the expressions “Unestablishable” in conclusions (6), (10) and (11) from y = aln(x) + b (b > 0, x > 0) were respectively revised for clearer texts as “No relationship between b1/b2 = MIC2/MIC1 can be established”, “No relationship for a1 + a2 can be established”, and “No correlation between the MPC and the |b| value was observed”.
(2) For section “2.2 Correlation between the MPSI and the MIC, MPC or SI ratio of two agents in an antimicrobial combination”
In the original manuscript, we had only calculated the MPSIs and the MPSIs (1:1) from those equations y = a1xb1 and y = a2xb2. Originally, these data, together with the tested MPSI and the tested MPSI (1:1), were intended to explore whether the MPSI (namely, the tested MPSI) can be replaced by the calculated MPSI (1:1) which is a simpler index. As these data are less and not good as a table, we put them with other data (including the MIC, MPC, and SI ratio values) together as Table 3 in the original manuscript.
According to your suggestion, we detached these data from Table 3, and presented them as Table S5 in the revised supplementary file, supplementing the calculated MPSIs and the calculated MPSIs (1:1) from those equations y = a1ln(x)+b1 and y = a2ln(x)+b2. In Table S5, the correlations between the tested MPSI and the calculated MPSI from y = axb or y = aln(x) + b, between the tested MPSI and the tested MPSI (1:1), and between the tested MPSI (1:1) and the calculated MPSI (1:1) from y = axb or y = aln(x) + b, were respectively established, and the significance was also analyzed using statistical r-test.
To achieved this, many revisions were made in the revised manuscript, including the adjustment of content and the revision of footnote d for Table 3, the deletion of some contents in paragraph 3, the rewriting of paragraph 7, and the insertion of paragraph 8. As paragraph 8 inserted may be closely related to statistical analyses you mentioned, here we presented it as follows for your consideration.
After the correlation analyses, it was surprisingly found that the MPSIs (1:1) (y) calculated from power function y= axb in Table 1 when x = 1 are approximately equal to the tested ones (x), with a linear equation y = 1.0398x and an r of 0.9972 (P < 0.01), and that there is significant (P < 0.01) correlation between the tested MPSIs (1:1) (y) and the tested MPSIs (x), with a linear equation y = 1.9392x and an r of 0.9672. These indicated that the tested or calculated MPSI (1:1) can replace the MPSI for effectively evaluating the potency difference preventing AMR of two agents in a drug combination, while the calculated MPSI (1:1) can be easily calculated after two equations y = a1xb1 and y = a2xb2 for two agents in a drug combination have been established using the data pair (1, a) and another, obtained from the experiment. Another, the significant correlation between the tested MPSI (x) and the calculated one (y), with a linear equation y = 1.5884x and an r of 0.9972 (P < 0.01), also indicated that the tested MPSI (namely, the MPSI in Table 3) can be roughly calculated from the established two equations y = a1xb1 and y = a2xb2 in a drug combination. However, five negative values were presented for nine MPSIs calculated from logarithmic function y = aln(x)+b in Table 1 when x = 1, and simultaneously it was found that there is no correlation between the tested MPSI (1:1) (x) and the calculated one (y) from logarithmic function, with a linear equation y = 0.4523x and an r of 0.4562 (P > 0.01). These further indicated power function y= axb is better than logarithmic one y = aln(x)+b to reflect the correlation between the SI value and the ratio value of two agents in a drug combination.
These above revisions for your consideration, and we are not sure whether these revisions gave the reasonable presentation for your suggestion.
Thank you very much for your valuable suggestion!
Point 3: Does the introduction provide sufficient background and include all relevant references? Can be improved
Response: Thank you for your valuable suggestion!
According to your suggestion, we had already improved the logic of the Introduction section, and highlighted the importance of the research topic and its applicability as possible as we can. For example,
- In first paragraph, we inserted the following text: Thereby, it is desperate to discover some regularity conclusions and put forward some proposals and applicable schemes, for effectively guiding the practices of drug combinations preventing AMR, simultaneously avoiding the accelerated spread of AMR due to the abuse of drug combinations.
- The last paragraph of the Introduction section was rewritten as follows: Here, based on our previous data [24], the deeper analyses were performed for dis-covering probable correlations between the various indexes (such as MIC, MPC, fractional inhibitory concentration index (FICI) and mutant selection index (SI)), predictors, and laws, and then the analysis conclusions were verified and improved through the communications with the practical effects preventing AMR of different proportional combinations after antibiotic exposure experiments. Based on this, a preliminary scheme that can guide the practice of drug combination preventing AMR, together with some proposals and laws, was put forward for further experimental improvements and clinical trials.
- We had provided more detailed background in the second paragraph of revised manuscript. The text of the second paragraph was presented as follows: To prevent AMR, the hypotheses of mutant selection window (MSW) and mutant prevention concentration (MPC) were put forward by Zhao and Drlica [17]. According to these hypotheses, maintaining drug concentrations above its MPC throughout therapy can severely restrict the acquisition of drug resistance. Based on these hypotheses, many related experiments have been performed for discovering probable parameters to predict drug combination effects on the prevention of AMR or the selection of resistant mutant, and several MPC- or MSW-related parameters were proposed [18-22], such as MPC level (AAMPC), AUC24/MPC (Area under the concentration-time curve over 24 h divided by the MPC) and Cmax/MPC (Highest concentration divided by the MPC), and AUC24/MIC (AUC24 divided by the minimal inhibitory concentration) [23]. These results indicated that anyone of those predictors cannot be widely used to guide the practices of drug combinations preventing AMR.
These above revisions for your consideration.
Thank you very much!
Point 4: Are all the cited references relevant to the research? Can be improved
Response: Thank you for your valuable suggestions!
According to your suggestion, we had already checked the cited references, taking a new reference to replace reference 5 in the original manuscript, and inserting three closely relevant references 14 to 16 in the revised manuscript. To improve the relevance of the cited references to the research, we had also adjusted the cited location of some references for more appropriate citations instead of relatively vaguer ones in the original manuscript, especially in the Introduction section.
These revisions had been performed for your consideration.
Thank you very much!
Other revisions:
Besides the Introduction section, we had carefully performed extensive revision throughout the manuscript including references, linguistic edit, expression, spelling, and grammar, etc. for your consideration.
Thank you very much!

Round 2
Reviewer 3 Report
Dear Authors,
Thank you very much for your prompt and extensive revision.
Author Response
Dear Academic Editor,
My co-authors and I are very grateful to you for your careful review, objective evaluations, kind reminders and valuable suggestions. We have amended the manuscript according to the issues raised by you, and have pleasure to submit the revised version, together with the response to all points, for your consideration.
Many thanks for your kind attention and help to improve our work!
Yours sincerely,
Ganjun Yuan
Here are our answers to your comments.
Unfortunately, while going through the pages, I started getting confused. I would like to suggest some clarifications and modifications which, in my opinion are necessary to understand what the present research adds to previous research done by your team, and to know the degree of generalization from your research.
Response: Thanks for your comments and constructive suggestions!
According to your suggestion, we had carefully revised the whole manuscript, especially sections “Abstract”, “Results”, “Discussions” and “Methods”. The details can be found in the following responses to your comments.
1) The abstract. It is quite confusing. After starting with a good background, I feel the text mixes previous research with the current research without clearly distinguishing both. For example: "To guide the practice, some regularity conclusions had been drawn in our previous work. Based on those experimental data, here the power function (y=axb, a > 0) correlation between the mutant selection index (SI) (y) of one agent and the ratio value (x) of two agents in a drug combination was established, and two rules a1 × MIC1 = a2 × MIC2 and b1 + b2 = -1 were discovered from both equations of y=a1xb1 and y=a2xb2 for two agents in drug combinations. Simultaneously, it was found that one agent with larger MPC alone for drug combinations would present a larger |b| and show greater potency for narrowing itself MSW and preventing the resistance."
This is just an example. An abstract is a clear, well-explained summary of the current research that, formally structured or not, starts with the background, defines the objective, summarises the method, and main findings and ends with the most important conclusion.
Response: Thank you for your objective comment, careful review and valuable suggestion!
According to your suggestion, we had clearly defined the related conclusions from our previous work, for distinguishing the current research from the previous research. Another, we adjusted the logic of the text throughout the abstract and revised some texts and expresses.
The manuscript involved the establishment of many novel correlations and laws, the proposals of two new concept, some principles to judge whether the resistance and the collateral sensitivity occur to two agents in drug combinations, and a preliminary scheme for antimicrobial combinations preventing the AMR. As they are interrelated for the contents, we don’t want to disassemble them to two or more manuscripts for maintaining the integrity of the research. However, this will increase the difficulty in writing, and it is difficult for us to write according to general writing mode although what you mentioned is right. For all that, we have been trying our best to improve the work according to the suggestions from you and subsequent other scholars, since we simultaneously put them as a preprint online for extensive discussion and improvement at any time.
These above revisions and interpretations were provided for your consideration.
2) The Methods section. This is crucial. An introductory sentence before starting with section 4.1. (Antimicrobial agents) is necessary. Which kind of study are you conducting? An overview sentence must precede the subsequent text. The idea of the detailed description of the methods is to give enough details to readers, to figure out how the study has been conducted; to provide enough elements to judge if the process follows the scientific methods; to identify possible gaps and biases and most importantly, to allow other researchers to repeat the research if they desire. None of this is facilitated in this section. Additionally, the Authors only describe Roxithromycin and Doxycycline, while the tables include Vancomycin, Ofloxacin and Fosfomycin. That incongruence cannot be accepted.
Response: Thank you for your kind reminder and valuable suggestion!
According to your suggestion, we inserted some content as an introduction of section 4. (Methods) before starting with section 4.1. (Antimicrobial agents), and more detailed description of the methods were provided for reader to easily allow other researchers to repeat the research if they desire.
According to the logic of this research, the subsection 4.3 should be arranged as subsection 4.1. However, the agents and materials used were generally presented as the first part in section Methods. So, we successively arranged the content “Antimicrobial agents” and “Isolates and media” as subsections 4.1 and 4.2, and which would probably lead readers to difficultly figure out how the study has been conducted. Thereby, we adjusted the arrangement of subsection, and moved subsection 4.3 to subsection 4.1, and respectively revised subsections 4.1, 4.2, 4.4 and 4.5 as subsections 4.2.1, 4.2.2, 4.2.3 and 4.2.4 of section 4.2. Simultaneously, some related information involving roxithromycin, doxycycline, vancomycin, ofloxacin and fosfomycin were inserted as the introductions of sections 4.1 and 4.2.
After these adjustments and revisions, the logic of this section corresponding to section 2. Results is clearer and more reasonable. This should allay your worries, and section 4. Methods should provide enough detail for repeating this research, especially the correlation analyses can be completely repeated.
Another, all related data of vancomycin, ofloxacin and fosfomycin, obtained from our previous work, were properly cited after above revisions. If other researchers desire to repeat the research, they can obtain the detail from section Methods in our previous paper [24]. As only combination roxithromycin/doxycycline was selected for exposure experiments in section 4.2, we only describe roxithromycin and doxycycline in section 4.2.1 (Antimicrobial agents), corresponding 4.1 (Antimicrobial agents) in the original manuscript.
These above revisions and interpretations were provided for your consideration.
3) The results section of a research article is intended to describe the results of THAT research without interpretations and references. Just results, data. Unfortunately, the Authors start mixing the results and conclusions obtained in previous research (Ref. 24) with the present research results. So, in the end, the reader gets confused, and it is impossible to clearly identify what is new and what is recycled material. Perhaps this is the reason why some tables include references to fixed-dose combinations (vanco/fosfo, for example), not even cited in the methods. This is also incorrect and confusing.
Response: Thank you for your kind reminder and valuable suggestion!
You are right! The results section of a research article is intended to describe the results, without interpretations and references. Thereby, we had performed extensive revisions for section 2. Results, and moved many texts and contents from this section to section 4. Methods. Since the manuscript is closely related to our previous work, a few citations for Ref. 24 sill remained, while some expressions were revised for avoiding to mix the results and conclusions obtained in previous research with the present research results.
Another, some related information of three combinations were inserted and properly cited in subsection 4.1 of the revised manuscript.
These above revisions and interpretations were provided for your consideration.
4) For me, at least, it is difficult to understand how "laws" and "rules" are reached out. In my opinion, researchers formulate hypotheses that, if confirmed with larger studies and by other authors, they become laws and rules. So, without explaining clearly how these conclusions are reached, it seems daring to refer to laws and rules without balancing these words with "suggested", "hypothesized", etc.
Response: Thank you for your kind reminder and valuable suggestion!
Inspired by your reminder, we searched the definitions of words "laws", "rules" and "principles" from Wikipedia, the free encyclopedia. From the results and corresponding methods, some results were obtained using mathematical statistics methods, especially for all the established correlations which can be completely repeated by other researchers, and some results were further verified by the antibiotics exposure experiments. Another, we also provided some supports from published papers reported by other authors in section 3. Discussions. So, we don’t think all the conclusions are just hypotheses although some principles and the preliminary scheme likely need to be confirmed with larger studies and by other authors. However, you are right! it seems daring to refer to laws and rules. Thereby, we used some words with "suggested", "proposed", etc. to balance it according to your suggestion.
These above revisions and interpretations were provided for your consideration.
5) And this leads me back to the title. In my opinion, an overpromising title that does not benefit the Authors because the reader starts reading with a perspective that is not fulfilled: "Drug combinations to prevent antimicrobial resistance: some correlations, rules and laws, and a preliminary scheme"
In fact, this is just a hypothesis (an interesting hypothesis, which could be very helpful for recommending or not hundreds of fixed-dose combinations being marketed in different countries). A more humble title, describing the reality of the study and the hypothesis suggested by the research, would clearly be more appropriate.
Response: Thank you for your comment and valuable suggestion!
As we responded to point 4), we don’t think this research is just a hypothesis although some principles and the preliminary scheme likely need to be confirmed with larger studies and by other authors. However, you are right! A more humble title, describing the reality of the study and the hypothesis suggested by the research, would clearly be more appropriate. So, we revised the title of the manuscript as “Drug combinations to prevent antimicrobial resistance: some correlations and laws, and their verifications, thus proposing some principles and a preliminary scheme” for your consideration.
Another, after further verified and improved by us and other researchers in the future, we hope that this research is not only helpful for recommending or not hundreds of fixed-dose combinations being marketed in different countries, but also is helpful for judging whether the clinical use of antibiotics is rational and selecting proper antibiotics to simultaneously obtain good clinical antimicrobial effects and AMR-preventing effects.
6) My recommendation in this opportunity is to rewrite the whole manuscript, keeping the references to the previous research to a minimum (because it has already been published), trying to avoid mixing the past data with the present data and trying to be clear in thinking about readers outside the laboratory.
Answer: Thank you for your valuable recommendation!
According to your suggestion, we had already performed extensive revisions, and kept the references to the previous research to a minimum. After revised, only a few citations for Ref. 24 carefully remained.
Thank you very much for your help to improve our work!
Other revisions
The Abstract graphic and Figure 3 were also revised for clearly presenting the logic of this research. Another, we had performed extensive revisions throughout the manuscript, including the check for references and supplementary file.
Thank you very much for your careful review, objective evaluations, kind reminders and valuable suggestions.
